# CONTEXTUAL MOLECULE REPRESENTATION LEARNING FROM CHEMICAL REACTION KNOWLEDGE

## ABSTRACT

In recent years, self-supervised learning has emerged as a powerful tool to harness abundant unlabelled data for representation learning, and has been broadly adopted in diverse areas. However, when applied to molecular representation learning (MRL), prevailing techniques such as masked sub-unit reconstruction often fall short, due to the high degree of freedom in the possible combinations of atoms within molecules, which brings insurmountable complexity to the masking-reconstruction paradigm. To tackle this challenge, we introduce *REMO*, a self-supervised learning framework that takes advantage of well-defined atom-combination rules in common chemistry. Specifically, *REMO* pre-trains graph/Transformer encoders on 1.7 million known chemical reactions in the literature. We propose two pre-training objectives: Masked Reaction Centre Reconstruction (MRCR) and Reaction Centre Identification (RCI). *REMO* offers a novel solution to MRL by exploiting the underlying shared patterns in chemical reactions as *context* for pre-training, which effectively infers meaningful representations of common chemistry knowledge. Such contextual representations can then be utilized to support diverse downstream molecular tasks with minimum finetuning, such as molecule property prediction and affinity prediction. Extensive experimental results on MoleculeNet, MoleculeACE, ACNet, and drug-drug interaction (DDI) show that across all tested downstream tasks, *REMO* outperforms the standard baseline of single-molecule masked modeling used in current MRL.

## 1 INTRODUCTION

Recently, there has been a surging interest in self-supervised molecule representation learning (MRL) (Rong et al., 2020; Guo et al., 2021; Xu et al., 2021; Lin et al., 2022), with aims at a pre-trained model learned from a large quantity of unlabelled data to support different molecular tasks, such as molecular property prediction (Wu et al., 2018), drug-drug interaction (Ryu et al., 2018), and molecule generation (Irwin et al., 2020). Though demonstrating promising results, these pre-training methods still exhibit critical drawbacks. For example, as illustrated in Figure 1b, two molecules that have significantly different chemical properties can exhibit very similar structure. This is called *activity cliff* (Stumpfe et al., 2019), a crucial differentiation task in drug discovery, where most neural networks (especially pre-trained models) struggle with poor performance not even comparable to a simple SVM method based on traditional fingerprint features (van Tilborg et al., 2022; Zhang et al., 2023).

Despite different architectures (transformer or graph neural networks), most current methods use a BERT-like masked reconstruction loss for training. This omits the high complexity of atom combinations within molecules in nature that is quite unique compared to a simple sentence comprised of a few words. To better capture the hidden intrinsic characteristics of molecule data, we need to look beneath the surface and understand how molecules are constructed biologically. For example, unlike the assumption in distributed semantic representation (Firth, 1957) that the meaning of a word is mainly determined by its context, in molecule graphs the relationships between adjacent sub-units are mostly irrelevant. Another main difference is that while changing one word in a long sentence might have a relatively low impact on the semantic meaning of the full sentence, in molecular structure one minor change might lead to a totally different chemical property. As illustrated in Figure 1a, given a pair of molecules with similar constituents, adding either a carbon or an oxygen atom are both reasonable choices in reconstructing it into a real molecule, but each resulting molecule exhibits

very different chemical properties. In such cases, traditional masked reconstruction loss is far from being sufficient as a learning objective.

Is there an essential natural principle that guides the atomic composition within molecules? In fact, most biochemical or physiological properties of a molecule are determined and demonstrated by its reaction relations to other substances (e.g., other small molecules, protein targets, bio-issues). Further, these natural relations between different reactants (e.g., molecules) are reflected by chemical reaction equations that are public knowledge in the literature. These chemical reaction relations offer a reliable context for the underlying makeup of molecule sub-structures, especially for reactive centres. As in Figure 1a, the choice of carbon or oxygen can be easily determined by different reactants, even though the two molecules have very similar structures.

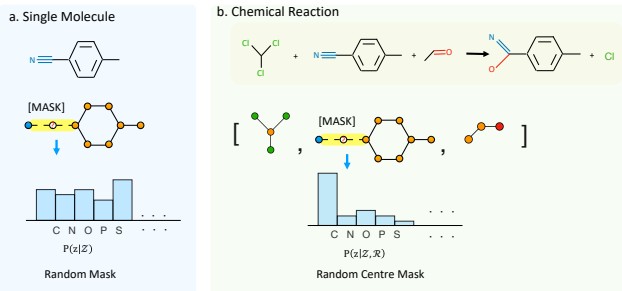

Figure 1: **a**: Adding either carbon or oxygen to the left molecule can lead to the formation of a valid molecule. However, these resulting molecules exhibit distinct reactions and consequently possess varying properties. **b**: Example of an activity cliff pair on the target Thrombin(F2), where $K_i$ is to measure the equilibrium binding affinity for a ligand that reduces the activity of its binding partner.

Figure 2: **a**: Traditional masked language model on a single model with objective $P(z|\mathcal{Z})$, where $z$ is the masked substructure and $\mathcal{Z}$ stands for the remaining ones. **b**: Our proposed Masked Reaction Centre Reconstruction with objective $P(z|\mathcal{Z}, \mathcal{R})$, where $\mathcal{R}$ denotes other reactants in chemical reactions.

Motivated by these insights, we propose a new self-supervised learning method, *REMO*, by harnessing chemical reaction knowledge to model molecular representations. Specifically, given a chemical reaction equation known in the literature, we mask the reaction centres and train the model to reconstruct the missing centres based on the given reactants as context. The key difference between *REMO* and existing methods that apply masked language model to a single molecule is the usage of *sub-molecule interaction context*, as illustrated in Fig 2. By relying on chemical reactants as context, the degree of freedom in possible combinations of atoms within a molecule is significantly reduced, and meaningful sub-structure semantics capturing intrinsic characteristics can be learned in the reconstruction process.

Extensive experiments demonstrate that *REMO* achieves new state-of-the-art results against other pre-training methods on a wide range of molecular tasks, such as activity-cliff, molecular property prediction, and drug-drug interaction. And to the best of our knowledge, *REMO* is the first deep learning model to outperform traditional methods on activity-cliff benchmarks such as MoleculeACE and ACNet.

## 2 RELATED WORK

### 2.1 SELF-SUPERVISED LEARNING FOR MOLECULE REPRESENTATION

Since this work is related to masked language model (MLM), we review recent MLM-based molecular representation learning methods. MLM is a highly effective proxy task, which has recently been applied to molecular pre-training on SMILES sequences, 2D graphs, and 3D conformations.

For pre-training on SMILES sequences, existing work including ChemBERTa (Chithrananda et al., 2020), MG-BERT (Zhang et al., 2021) and Molformer (Ross et al., 2022) directly apply MLM to SMILES sequence data, and conduct similar reconstruction optimization to obtain molecular representations. When applied to 2D graphs, different masked strategies have been proposed. For example, AttrMask (Hu et al., 2020) randomly masks some atoms of the input graph and trains a Graph Neural Network (GNN) model to predict the masked atom types. GROVER (Rong et al., 2020) trains the model to recover masked sub-graph, which consists of k-hop neighbouring nodes and edges of the target node. Mole-BERT (Xia et al., 2023) uses a VQ-VAE tokenizer to encode masked atoms with contextual information, thereby increasing vocabulary size and alleviating the unbalanced problem in mask prediction. By capturing topological information in a graph, these methods have the ability to outperform the aforementioned SMILES-based methods.

Recently studies also apply MLM to 3D conformations. For example, Uni-MOL (Zhou et al., 2023) predicts the masked atom in a noisy conformation, while ChemRL-GEM (Fang et al., 2022) predicts the atom distance and bond angles from masked conformation. Yet these methods suffer from limited 3D conformation data.

In summary, MLM has been widely used in molecule representation over different data forms (1D, 2D, 3D). Given a single molecule, some sub-structures are masked, and a reconstruction loss is computed to recover the semantic relations between masked sub-structures and remaining contexts. However, this training objective does not fit the essence of molecule data and may lead to poor performance. For example, MoleculeACE (van Tilborg et al., 2022) and ACNet (Zhang et al., 2023) demonstrate that existing pre-trained models are incomparable to SVM based on fingerprint on activity cliff.

### 2.2 CHEMICAL REACTION MODELS

Most existing work on modelling chemical reactions is based on supervised learning (Jin et al., 2017; Coley et al., 2019). For example, Jin et al. (2017) and Coley et al. (2019) use a multiple-step strategy to predict products given reactants, and achieves human-expert comparable accuracy without using reaction templates.

Recently, self-supervised learning methods have been proposed to utilize chemical reaction data for pre-training, including Wen et al. (2022), Wang et al. (2022) and Broberg et al. (2022). Wen et al. (2022) conducts a contrastive pre-training on unlabelled reaction data to improve the results of chemical reaction classification. Wang et al. (2022) focuses on introducing reaction constraints on GNN-based molecular representations, to preserve the equivalence of molecules in the embedding space. Broberg et al. (2022) pre-trains a transformer with a sequence-to-sequence architecture on chemical reaction data, to improve the performance of molecular property prediction.

Although these methods use chemical reaction data for pre-training, *REMO* is different as the objective is to focus on rection centre, i.e. masked rection centre construction and identification, which better fits chemical reaction data. *REMO* also provides a general framework that can be used in different architectures, such as transformer and GNNs, to support diverse downstream tasks.

## 3 REMO: CONTEXTUAL MOLECULAR MODELING

This section describes the molecule encoder and the pre-training process of *REMO*, specifically, its two pre-training tasks: Masked Reaction Centre Reconstruction (*MRCR*) and Reaction Centre Identification (*RCI*).

### 3.1 MOLECULE ENCODER

We design *REMO* as a self-supervised learning framework, which is agnostic to the choice of encoder architecture. In this work, we represent each molecule as a 2D graph $G = (V, E)$, where $V$ is the set of atoms and $E$ is the set of bonds, and each bond or node is represented as a *one-hot*

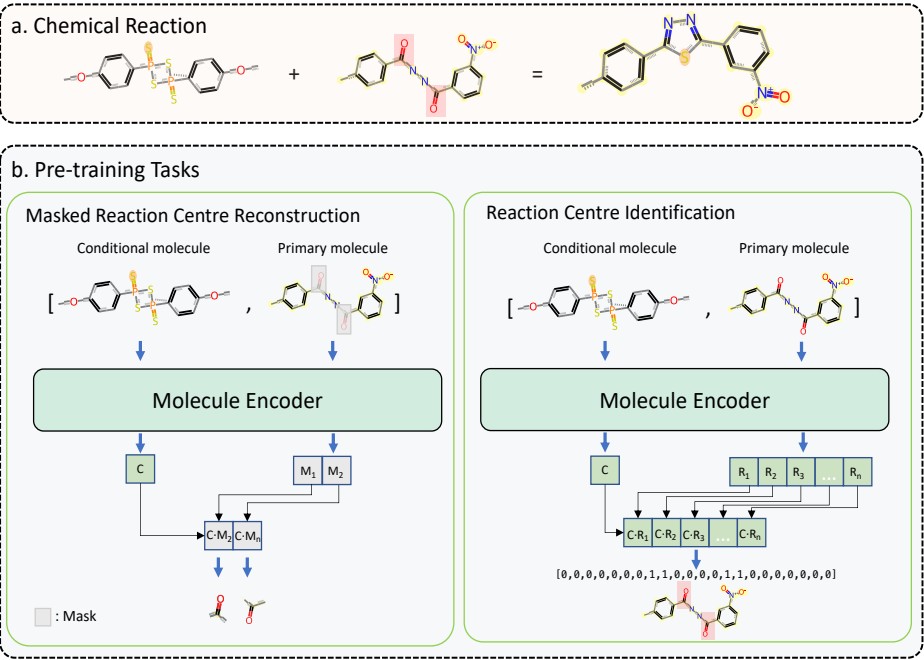

Figure 3: Architecture of *REMO*. **a**: An example chemical reaction consisting of two reactants and one product, where the primary reactant has two reaction centres. **b**: Illustrations of two pre-training tasks in *REMO*, *i.e.*, Masked Reaction Centre Reconstruction and Reaction Centre Identification.

*initialized embedding* based on its type. Particularly, we experiment with two model architectures for pre-training: one is GNN-based, with Graph Isomorphism Network (GIN) (Xu et al., 2018) as representative; the other is transformer-based, with Graphormer as the backbone (Ying et al., 2021).

GIN (Xu et al., 2018) learns representations for each node and the whole graph through a neighbour aggregation strategy. Specifically, each node $v \in V$ updates its representation based on its neighbours and itself in the aggregation step, and the $l$-th layer of a GIN is defined as:

$$h_v^{(l)} = \text{MLP}^{(l)} \left( \left( 1 + \epsilon^{(l)} \right) \cdot h_v^{(l-1)} + \sum_{u \in \mathcal{N}(v)} h_u^{(l-1)} + \sum_{u \in \mathcal{N}(v)} e_{uv} \right) \quad (1)$$

where $h_v^{(l)}$ is the representation of node $v$ in the $l$-th layer, $e_{uv}$ is the representation of the edge between $u$ and $v$, and $\mathcal{N}(v)$ stands for the neighbors of $v$.

Then, we obtain the graph-level representation through a "READOUT" function, with the average of node features used as the pooling layer:

$$h_G = \text{READOUT} \left( \left\{ h_v^{(K)} \mid v \in G \right\} \right). \quad (2)$$

Graphormer (Ying et al., 2021) is the first deep learning model built upon a standard Transformer that greatly outperforms all conventional GNNs on graph-level prediction tasks, and has won the first place in the KDD Cup-OGB-LSC quantum chemistry track (Hu et al., 2021). The basic idea is to encode graph structural information to positional embeddings, to facilitate Transformer architecture. Specifically, Graphormer introduces three types of positional embeddings, and we simply follow the original architecture. The attention in Graphormer is calculated by:

$$A_{ij} = \frac{(h_i W_Q) (h_j W_K)^T}{\sqrt{d}} + b_{\phi(v_i, v_j)} + c_{ij}, \quad c_{ij} = \frac{1}{N} \sum_{n=1}^{N} x_{e_n} \left( w_n^E \right)^T, \quad (3)$$

where $h_i$ and $h_j$ are centrality embeddings to capture the node importance in the graph, $b_{\phi(v_i, v_j)}$ is spatial encoding calculated by the shortest path (SP) to capture the structural relation between nodes,

and $c_{ij}$ is the edge embedding. $x_{en}$ is the feature of the $n$-th edge $e_n$ in $\text{SP}_{ij}$, $w_n^E \in \mathbb{R}^{d_E}$ is the $n$-th weight embedding, and $d_E$ is the dimensionality of edge feature. Apart from adjustments on the attention mechanism, Graphormer follows the original Transformer (Vaswani et al., 2017) to obtain embeddings.

## 3.2 LEARNING OBJECTIVE

Our design philosophy of the learning objective is as follows: rection centres are the core of a chemical reaction, which depict meaningful structural and activity-related relations within a molecule. Specifically, chemical reaction describes a process in which one or more substances (*i.e.*, reactants) are transformed to one or more different substances (*i.e.*, products). This process involves exchanging electrons between or within molecules, and breaking old chemical bonds while forming new ones. It has been observed that only parts of molecules are directly involved in reactions (known as 'reaction centres') in organic chemistry. Therefore, rection centres are the key to revealing the behaviors of molecule reactivity properties during the chemical reaction process.

Formally, each chemical reaction is viewed as two graphs: reactant graph $G_r$ and product graph $G_p$, which are unconnected when there are multiple reactants or products. Since reaction centres are usually not provided by chemical reaction data, we utilize a widely used technique (Jin et al., 2017; Shi et al., 2020) to detect corresponding reaction centres. Specifically, the reaction centre for a chemical reaction process is represented as a set of atom pairs $(a_i, a_j)$, where the bond type between $a_i$ and $a_j$ differs between the aligned reactant graph $G_r$ and the product graph $G_p$. Intrinsically, the reaction centre is the minimum set of graph edits to transform reactants to products.

## 3.3 MASKED REACTION CENTRE RECONSTRUCTION

We propose a new pre-training task, where each reactant in a chemical reaction is treated as the primary molecule, and the remaining reactants are treated as its contexts or conditional molecules. The reaction centre atoms in the primary molecule and their adjacent bonds are masked with a special token, denoted as $[MASK]$. Then all the conditional molecules are fed into the molecule encoder to obtain a global representation. Specifically for GIN encoder, the node vector of each atom in conditional molecules are pooled to produce a global representation $c$. While for Graphormer, the process is a bit different. To adapt to the transformer architecture, we introduce a virtual node to connect all the atoms together, and the representation of the virtual node in the last layer is treated as the global representation.

The masked primary molecule with $k$ reaction centres is directly fed into the molecule encoder (GIN or Graphormer), and the final hidden vectors corresponding to each $[MASK]$ token are produced by concatenating the vectors of atoms belonging to this reaction centre, denoted as $M_j, \forall j = 1, \cdots, k$.

After that, both $c$ and each $M_j, j = 1, \cdots, k$ are fed into a 2-layer MLP to output a softmax over the vocabulary to predict the probability of the masked reaction centre. Specifically, the vocabulary is produced by enumerating all the topological combinations of the atoms and their adjacent bonds. The model is expected to predict the exact atom type and the adjacent bonds during the pre-training stage:

$$P(g_j) = \texttt{softmax}(MLP(c, M_j)), \forall j = 1, \cdots, k, \tag{4}$$

where $g_j$ stands for the graph representation of the $j$-th reaction centre, including both atom types and adjacent bonds. Finally, the sum of negative likelihood loss is used for training.

## 3.4 REACTION CENTRE IDENTIFICATION

Another novel pre-training objective in *REMO* directly conducts classification on each atom without introducing any $[MASK]$ label. For each primary molecule in a chemical reaction, all the conditional molecules are fed into the molecule encoder to obtain a global representation $c$, similar to that in Masked Reaction Centre Reconstruction. The primary molecule is directly fed into the molecule encoder (GIN or Graphormer), and the final hidden vectors corresponding to each atom are produced, denoted as $R_j, \forall j = 1, \cdots, N$, where $N$ stands for the number of atoms in the primary molecule.

After that, both $c$ and each $R_j, j = 1, \cdots, N$ are fed into a 2-layer MLP to predict whether an atom belongs to a reaction centre, as follows:

$$P(y_j) = \texttt{softmax}(MLP(c, R_j)), \forall j = 1, \cdots, N, \tag{5}$$

where $y_j = 1/0$ denotes whether the $j$-th atom is a reaction centre. During training, the sum of the negative likelihood loss is utilized as the objective function.

|  |  | $RMSE$ | $RMSE_{cliff}$ |
|---|---|---|---|
| Fingerprint based | ECFP+MLP | 0.772 | 0.831 |
|  | ECFP+SVM | **0.675** | **0.762** |
| SMILES based | SMILES+Transformer | 0.868 | 0.953 |
|  | SMILES+CNN | 0.937 | 0.962 |
|  | SMILES+LSTM | 0.744 | 0.881 |
| Graph based | GAT | 1.050 | 1.060 |
|  | GCN | 1.010 | 1.042 |
|  | AFP | 0.970 | 1.000 |
|  | MPNN | 0.960 | 0.998 |
|  | GIN (Xu et al., 2018) | 0.815 | 0.901 |
|  | Graphormer (Ying et al., 2021) | 0.805 | 0.848 |
| Pre-training based | AttrMask-GIN (Hu et al., 2020) | 0.820 | 0.930 |
| Ours | $REMO_M$-GIN | 0.807 | 0.922 |
|  | $REMO_I$-GIN | 0.767 | 0.878 |
|  | $REMO_M$-Graphormer | 0.679 | 0.764 |
|  | $REMO_I$-Graphormer | **0.675** | 0.768 |

Table 1: Comparison between *REMO* and baselines on MoleculeACE. The best and second best results are indicated in bold and underlined, respectively.

## 4 EXPERIMENTS

We evaluate *REMO* against state-of-the-art (SOTA) baselines on multiple molecular tasks, including activity cliff prediction, molecular property predicition, and drug-drug interaction. Extensive experiments show that *REMO* achieves new SOTA on all tasks.

### 4.1 IMPLEMENTATION

Firstly, we introduce the pre-training of *REMO*. We utilize USPTO (Lowe, 2017) as our pre-training data, which is a subset and preprocessed version of Chemical reactions from US patents (1976-Sep2016). From different versions of USPTO, we select the largest one, i.e. USPTO-full, which contains 1.9 million chemical reactions. After removing the reactions that fail to locate reaction centres, there remain 1,721,648 reactions in total, which constitutes our pre-training dataset. In statistics, this dataset covers 0.7 million distinct molecules as reactants.

We implement three versions of *REMO*, denoted as $REMO_M$, $REMO_I$ and $REMO_{IM}$, to represent using different training objectives, i.e. masked reaction centre reconstruction, reaction centre identification, and both. As for the molecule encoders, we implement both GIN and Graphormer, with the same configuration from the original paper. Specially for Graphormer, we select the small configuration. We apply Adam optimizer throughout the pre-training and fine-tuning procedure. Specifically, the learning rate is set to 3e-4, and the batch size is set to 128. We randomly split 10,000 reactions as the validation set, and the pre-training lasts for 5 epochs over the dataset.

### 4.2 ACTIVITY CLIFF PREDICTION

Activity cliff is a known challenging task, because all given molecule pairs are structurally similar but exhibit significantly different activities. An accurate activity cliff predictor requires to be sensitive to the structure-activity landscape. SOTA deep learning based molecular representation learning methods could not yield better results compare to fingerprint based methods on this task (van Tilborg et al., 2022; Zhang et al., 2023). In our experiments, we use two benchmarks, MoleculeACE and ACNet.

#### 4.2.1 EVALUATION ON MOLECULEACE

**Data** MoleculeACE (van Tilborg et al., 2022) includes 30 tasks, where each task provides binding affinities of different molecules to a specific protein target. Among these molecules, there are some activity-cliff pairs, spreading to training set and test set. The size of every single benchmark ranges from 615 to 3,658, and the range of cliff ratio (the percentage of activity cliff molecules to all molecules) is from 9% to 52%. In experiments, we follow the train-test split of the default setting of MoleculeACE, and accordingly stratified split the training set into train and validation sets by ratio 4:1.

| Models | AUC | Models | AUC |
|---|---|---|---|
| GROVER (Rong et al., 2020) | 0.753(0.010) | ChemBERT (Guo et al., 2021) | 0.656(0.029) |
| MAT (Maziarka et al., 2020) | 0.730(0.069) | Pretrain8 (Zhang et al., 2022) | 0.782(0.031) |
| pretrainGNNs (Hu et al., 2020) | 0.758(0.015) | S.T. (Honda et al., 2019) | 0.822(0.022) |
| GraphLoG (Xu et al., 2021) | 0.752(0.040) | GraphMVP (Liu et al., 2021) | 0.724(0.026) |
| ECFP | 0.813(0.024) | *REMO* | **0.828(0.020)** |

Table 2: Comparison between *REMO* and baselines on the *Few* subset of ACNet, with AUC-ROC(%) as the evaluation metric.

**Baselines** We compare with the baselines tested by MoleculeACE (van Tilborg et al., 2022), including fingerprint-based methods, SMILES-based methods and graph-based methods. Since we also use Graphormer as our molecule encoder, we implement its supervised version for comparison. In addition, we compare with a pre-training method, AttrMask (Hu et al., 2020) with GIN as the backbone.

**Finetuning** The pre-trained model of *REMO* is followed by a 2-layer MLP to obtain the logits. Since binding-affinity prediction is a regression task, we utilize RMSE as the loss for finetuning. For evaluation, we follow the default setting of MoleculeACE and report two metrics, averaged RMSE over the entire test set and averaged RMSE of the cliff molecules subset, denoted as $\text{RMSE}_{\text{cliff}}$.

**Results** Table 1 shows that: (1) Our contextual pretraing strategy is much more suitable for activity cliff, as compared to traditional masked strategy on single molecules. For example, the Attr-Mask model, a pre-trained GIN (0.820/0.930), performs worse than the original GIN (0.815/0.901). While if we pre-train GIN within *REMO*, better results (0.767/0.878) are achieved, as shown in $REMO_\text{I}$. (2) None of the deep learning baselines is comparable to the fingerprint-based method ECFP+SVM (0.675/0.762), while *REMO* has the ability to achieve comparable best results, by employing Graphormer as molecule encoder, as shown in $REMO_\text{I}$-Graphormer (0.675/0.768) and $REMO_\text{M}$-Graphormer (0.679/0.764).

### 4.2.2 EVALUATION ON ACNET

**Data** ACNet (Zhang et al., 2023) serves as another benchmark for activity cliff evaluation. Different from MoleculeACE, ACNet specifically focuses on classification task of determining whether a given molecular pair qualifies as an activity cliff pair. In our experiment, we employ a subset of tasks known as *Few* for evaluation, which is comprised of 13 distinct tasks with a total of 835 samples. Due to the limited size of training data, learning a model from scratch is challenging. Therefore, it is especially suitable to evaluate the transfer learning capability of pre-trained models. We adhere to the split method in ACNet, dividing each task into separate train, validation, and test sets.

**Baselines** We use the same baselines as the original ACNet, including self-supervised pre-trained methods such as Grover (Rong et al., 2020), ChemBERT (Guo et al., 2021), MAT (Maziarka et al., 2020), Pretrain8 (Zhang et al., 2022), PretrainGNNs (Hu et al., 2020), S.T. (Honda et al., 2019), GraphLoG (Xu et al., 2021), and GraphMVP (Liu et al., 2021). Additionally, we include another baseline method that utilizes ECFP fingerprint-based MLP for training. All the results here are from the original ACNet paper.

**Finetuning** Following the practice of ACNet, we extract the embedding of each molecule in the input pair with pre-trained *REMO*. Since Graphormer demonstrates superior performance to GIN, we only evaluate *REMO* with Graphormer as the molecule encoder hereafter. We concatenate these embeddings and feed them into a 2-layer MLP model to predict whether the pair exhibits an activity cliff. Each experiment is repeated three times, and the mean and deviation values are reported. Prediction accuracy is measured by ROC-AUC(%). We run finetuning with 3 random seeds, and take the average along with standard deviation (reported in bracket).

**Results** Table 2 shows that *REMO* clearly outperforms all other pre-trained methods, yielding the best results. This further demonstrates *REMO*'s proficiency in capturing the Quantitative Structure-Activity Relationship (QSAR).

### 4.3 MOLECULAR PROPERTY PREDICTION

**Data** We evaluate our method on MoleculeNet (Wu et al., 2018) which is a prevalent benchmark for molecular property prediction, including physiological properties such as BBBP, Sider, ClinTox,

| Dataset | BBBP | Tox21 | MUV | BACE | ToxCast | SIDER | ClinTox | HIV | Avg. |
|---|---|---|---|---|---|---|---|---|---|
| Graphormer-s | 67.1(2.0) | 73.7(0.4) | 58.3(3.2) | 70.4(3.9) | 65.2(0.5) | 63.4(0.5) | 72.5(0.3) | 71.7(0.5) | 67.8 |
| GROVER$_{base}$ (Rong et al., 2020) | 70.0(0.1) | 74.3(0.1) | 67.3(1.8) | 82.6(0.7) | 65.4(0.4) | 64.8(0.6) | 81.2(3.0) | 62.5(0.9) | 71.0 |
| GROVER$_{large}$ (Rong et al., 2020) | 69.5(0.1) | 73.5(0.1) | 67.3(1.8) | 81.0(1.4) | 65.3(0.5) | 65.4(0.1) | 76.2(3.7) | 68.2(1.1) | 70.8 |
| AttrMask (Hu et al., 2020) | 64.3(2.8) | 76.7(0.4) | 74.7(1.4) | 79.3(1.6) | 64.2(0.5) | 61.0(0.7) | 71.8(4.1) | 77.2(1.1) | 71.2 |
| GraphMAE (Hou et al., 2022) | 72.0(0.6) | 75.5(0.6) | 76.3(2.4) | 83.1(0.9) | 64.1(0.3) | 60.3(1.1) | 82.3(1.2) | 77.2(0.1) | 73.9 |
| GraphLoG (Xu et al., 2021) | 72.5(0.8) | 75.7(0.5) | 76.0(1.1) | 83.5(1.2) | 63.5(0.7) | 61.2(1.1) | 76.7(3.3) | 77.8(0.8) | 73.4 |
| Mole-BERT (Xia et al., 2023) | 72.3(0.7) | 77.1(0.4) | 78.3(1.2) | 81.4(1.0) | 64.5(0.4) | 62.2(0.8) | 80.1(3.6) | 78.6(0.7) | 74.3 |
| GraphMVP (Liu et al., 2021) | 68.5(0.2) | 74.5(0.4) | 75.0(1.4) | 76.8(1.1) | 62.7(0.1) | 62.3(1.6) | 79.0(2.5) | 74.8(1.4) | 71.7 |
| REMO$_I$ | **73.4(1.4)** | **78.5(0.7)** | 76.4(0.9) | 84.4(0.6) | 68.1(0.2) | **67.0(1.1)** | 64.6(6.0) | 77.3(1.5) | 73.7 |
| REMO$_M$ | 66.8(2.3) | 77.4(0.3) | 77.5(3.1) | 82.0(2.0) | 67.5(0.3) | 65.8(1.8) | 70.1(1.6) | 79.6(0.6) | 73.3 |
| REMO$_{IM}$ | 70.1(1.2) | **78.5(0.9)** | 77.4(2.1) | **84.5(0.4)** | **69.2(0.6)** | 65.4(0.5) | 75.2(5.4) | 77.4(1.3) | 74.7 |
| REMO$_{IM}$-AttrMask | 72.4(0.6) | 76.4(0.2) | 72.1(4.4) | 78.3(0.5) | 62.8(0.1) | 60.5(0.5) | 81.3(0.2) | 77.0(0.1) | 72.6 |
| REMO$_{IM}$-GraphMAE | 71.6(0.1) | 75.8(0.1) | 77.1(2.0) | 82.1(0.2) | 64.0(0.1) | 64.3(0.2) | **85.6(1.9)** | **79.8(0.1)** | 75.0 |

Table 3: Comparison between *REMO* and baselines on 8 classification tasks of MoleculeNet.

| Method | Accuracy | Precision | Recall | F1-Score |
|---|---|---|---|---|
| DeepDDI( (Ryu et al., 2018)) | 0.877 | 0.799 | 0.759 | 0.766 |
| DeepWalk( (Perozzi et al., 2014)) | 0.800 | 0.822 | 0.710 | 0.747 |
| LINE( (Tang et al., 2015)) | 0.751 | 0.687 | 0.545 | 0.580 |
| MUFFIN( (Chen et al., 2021)) | 0.939 | 0.926 | 0.908 | 0.911 |
| *REMO* | **0.953** | **0.932** | **0.932** | **0.928** |

Table 4: Evaluation of *REMO* on Drug-Drug Interaction task.

Tox21, and Toxcast, as well as biophysical properties such as Bace, HIV, and MUV. We use scaffold method to split each dataset into training, validation and test sets by ratio 8:1:1.

**Baselines** We compare our methods with various state-of-the-art self-supervised techniques that are also based on 2D graph, including AttrMask (Hu et al., 2020), GraphMAE (Hou et al., 2022), GraphLoG (Xu et al., 2021), Grover (Rong et al., 2020), Mole-BERT (Xia et al., 2023) and Graph-MVP (Liu et al., 2021). Since USPTO only covers 0.7 million molecules, we propose to continuously pre-train AttrMask and GraphMAE with both masked reaction centre reconstruction and identification objectives, denoted as *REMO*$_{IM}$-AttrMask and *REMO*$_{IM}$-GraphMAE, respectively.

**Finetuning** We append a 2-layer MLP after the pre-trained model to get the logits of property predictions, and minimize binary cross-entropy loss for the classification tasks during fine-tuning. ROC-AUC(%) is employed as evaluation metric for these classification tasks. All methods are run 3 times with different seed and the mean and deviation value are reported.

**Results** Table 3 shows that: (1) both pre-training objectives in *REMO* significantly improve the prediction performances, i.e. increasing from 67.8% (supervised Graphormer denoted as Graphormer-s) to 73.3% (*REMO*$_M$) and 73.7% (*REMO*$_I$), respectively. (2) By using only 0.7 million molecules for pre-training, *REMO* outperforms other methods such as MoleBERT and Grover which uses much larger data (i.e. 2 and 10 million molecules respectively), demonstrating the effectiveness of our contextual pre-training strategy. (3) *REMO*'s performance can be further improved when combined with other methods. As compared to the original AttrMask and GraphMAE, *REMO* improves their results from 71.2%/73.9% to 72.6%/75.0%, respectively.

## 4.4 DRUG-DRUG INTERACTION

**Data** We collect the multi-class drug-drug interaction (DDI) data following the work of MUFFIN (Chen et al., 2021). The dataset, which contains 172,426 DDI pairs with 81 relation types, is filtered from DrugBank (Wishart et al., 2018) that relationships of less than 10 samples are eliminated. We randomly split the dataset into training, validation and test set by 6:2:2.

**Baselines** To examine the effectiveness of our model, we compare REMO with several baselines, including well know grah based methods DeepWalk (Perozzi et al., 2014), LINE (Tang et al., 2015) and MUFFIN, and a typical deep learning based method DeepDDI (Ryu et al., 2018).

**Finetuning** We concatenate the paired global embedding of each molecule and feed it into a 2-layer MLP. Then a cross-entropy loss is utilized for fine-tuning. Typical classification metrics such as Precision, Recall, F1, and Accuracy are used for evaluation.

**Results** As shown in Table 4, our pre-trained model outperforms all the baselines, indicating that *REMO* effectively captures molecule interaction information from chemical reaction data.

## 4.5 ABLATION STUDIES

In order to further explain the advantage of *REMO* compared to traditional masked language model on single molecule, we conduct an ablation study to compare AttrMask and *REMO*, where Graphormer is utilized as backbones of both methods to ensure a fair comparison.

Firstly, we show the comparison results on MoleculeACE. As shown in Figure 4 Right, even though pre-trained Graphormer (i.e. Graphormer-p) outperforms supervised Graphormer (i.e. Graphormer-s), it is inferior to *REMO*, demonstrating the superiority of our contextual pre-training objective.

To explain why our pre-training strategy is much more effective, we propose to compare the information entropy. The main difference of our pre-trained model and the traditional one is that we utilize different context, i.e. we further use other reactants in chemical reactions as the context to determine the semantic of a substructure in a molecule. That is to say, we are modelling a conditional probability $P = P(z|\mathcal{Z}, \mathcal{R})$ as compared with the traditional one $Q = P(z|\mathcal{Z})$, where $\mathcal{Z}$ stands for the remaining substructures in a molecule and $\mathcal{R}$ stands for other reactants in the chemical reactions. It is clear that the information entropy in our case is much lower than that in traditional case, i.e. $H(P) \leq H(Q)$, meaning that the choice of masked sub-units becomes more definite under the context of chemical reactions.

To qualitatively evaluate their difference, we select 10,000 reactions from USPTO-full and conduct the reconstructions separately. The result in Figure 4 Left show a clear distribution shifts to left with the average information entropy dropping from 2.03 to 1.29, indicating that introducing reaction information contributes to the reaction centre reconstruction.

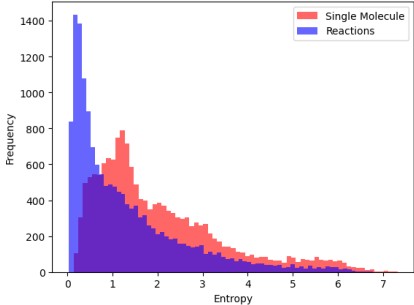

|  | RMSE | $RMSE_{cliff}$ |
| --- | --- | --- |
| Graphormer-s | 0.805(0.017) | 0.848(0.003) |
| Graphormer-p | 0.695(0.004) | 0.821(0.004) |
| *REMO*$_M$ | 0.694(0.006) | 0.771(0.004) |

Figure 4: *Left*: The information entropy of reconstructing reaction centres from reactions (*blue*), and reconstructing the same atoms of reaction centres without the context of reactions (*red*). *Right*: Ablation study of *REMO* on MoleculeACE.

## 5 CONCLUSION

We propose *REMO*, a novel self-supervised molecular representation learning method (MRL) that leverage the unique characteristics of chemical reactions as knowledge context for pre-training. By involving chemical reactants as context, we use both masked reaction centre reconstruction and reaction centre identification objectives for pre-training, which reduces the degree of freedom in possible combinations of atoms in a molecule as in traditional MLM approaches for MRL. Extensive experimental results reveal that *REMO* achieves state-of-the-art performance with less pre-training data on a varity of downstreaming tasks, including activity cliff prediction, molecular property prediction, and drug-drug interaction, demonstrating the superiority of contextual modelling with chemical reactions.

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

## A  MATHEMATICAL FORM OF THE ARCHITECTURES PRESENTED IN SECTION 3

The probability of pre-training from single molecules could be presented as

$$
\begin{aligned}
\mathcal{Q} = P(z|\mathcal{Z}) &= \Sigma_{z'} P(z|z', \mathcal{Z}) \cdot P(z'|\mathcal{Z}) \\
&= P(z|z'(\mathcal{Z}), \mathcal{Z})
\end{aligned}
\tag{6}
$$

where $z$ stands for the masked sub-units of molecules for reconstruction, $\mathcal{Z}$ stands for the remaining substructures in a molecule, $z'$ stands for the embedding of $z$ for prediction. $z'(\mathcal{Z})$ stands for $z'$ is solely dependent on $\mathcal{Z}$.

The probability of pre-training from chemical reactions, as *REMO* proposes could be presented as

$$
\begin{aligned}
\mathcal{P} = P(z|\mathcal{Z}, \mathcal{R}) &= \Sigma_{(\mathcal{R}', z')} P(z|z', \mathcal{R}', \mathcal{Z}, \mathcal{R}) \cdot P(z', \mathcal{R}'|\mathcal{Z}, \mathcal{R}) \\
&= P(z|z'(\mathcal{Z}), \mathcal{R}'(\mathcal{R}), \mathcal{Z}, \mathcal{R})
\end{aligned}
\tag{7}
$$

where $\mathcal{R}$ stands for the reactants other than $\mathcal{Z}$ in the chemical reaction, and $\mathcal{R}'$ stands for the embedding of $\mathcal{R}$. $\mathcal{R}'(\mathcal{R})$ stands for the embedding $\mathcal{R}'$ is solely dependent on $\mathcal{R}$.

## B  PROOF FOR THE SUPERIORITY OF REMO AT PRE-TRAINING PHASE

In order for a given molecule $\mathcal{Z}$ to undergo a reaction with another reactant $\mathcal{R}$, only specific sub-units $z$ can enable the occurrence of the reaction. It stands for the information gain between $z$ and $\mathcal{R}$ on the condition of $\mathcal{Z}$ is greater than 0.

$$
I(z, \mathcal{R}|\mathcal{Z}) > 0 \tag{8}
$$
$$
H(z|\mathcal{Z}) - H(z|\mathcal{Z}, \mathcal{R}) > 0 \tag{9}
$$
$$
H(z|\mathcal{Z}, \mathcal{R}) < H(z|\mathcal{Z}) \tag{10}
$$

Comparing to the reconstruction of $z$ solely conditioned on $\mathcal{Z}$, reconstructing $z$ as the reaction centre conditioned on both $\mathcal{Z}$ and $\mathcal{R}$ is more definite.

## C  DETAILED RESULTS ANALYSIS ON MOLECULEACE

### C.1  DETAILED RESULTS OF SOTA RESULTS ON MOLECULEACE

We show the detailed performance of 3 best methods including ECFP+SVM, *REMO*$_\text{I}$-Graphormer and *REMO*$_\text{M}$-Graphormer on 30 sub-tasks of MoleculeACE in Figure 5 and Figure 6, illustrating REMO's mastery in grasping the Quantitative Structure-Activity Relationship (QSAR).

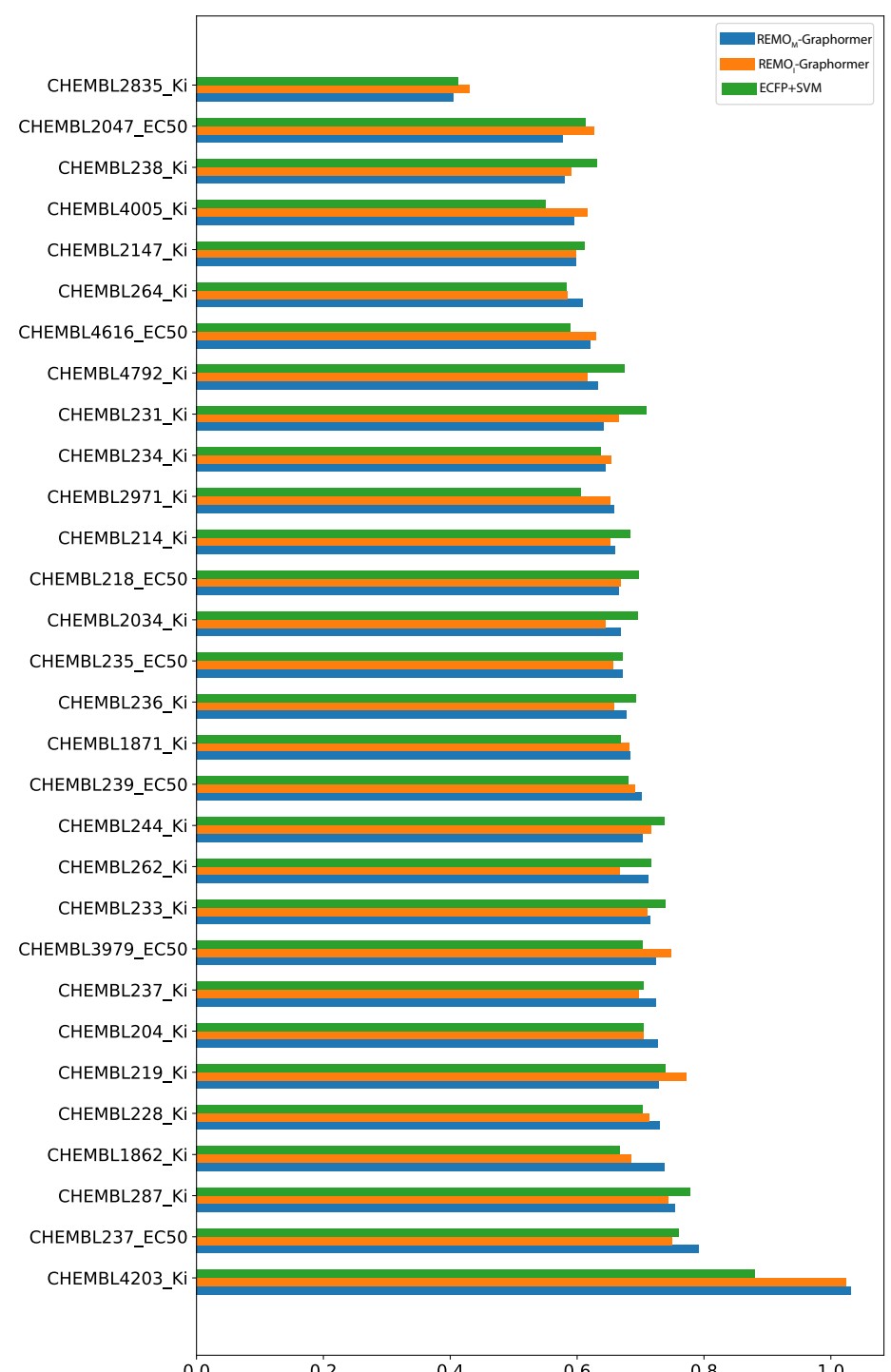

Figure 5: RMSE of $REMO_M$-Graphormer, $REMO_I$-Graphormer and ECFP+SVM on MoleculeACE

## C.2 DETAILED RESULTS OF ABLATION STUDIES ON MOLECULEACE

The detailed results of the conducted ablation studies on MoleculeACE are depicted in Figure 7 and Figure 8, showcasing the superiority of our contextual pre-training objective over the single molecular-based masking strategy.

| Dataset | BBBP | Tox21 | MUV | BACE | ToxCast | SIDER | ClinTox | HIV | Avg. |
|---|---|---|---|---|---|---|---|---|---|
| AttrMASK | 64.3(2.8) | 76.7(0.4) | 74.7(1.4) | 79.3(1.6) | 64.2(0.5) | 61.0(0.7) | 71.8(4.1) | 77.2(1.1) | 71.2 |
| AttrMASK - USPTO | 66.2(1.8) | 76.6(0.3) | 75.3(0.4) | 79.8(0.5) | 64.0(0.4) | 59.5(0.5) | 66.1(2.7) | 78.2(1.2) | 70.71 |
| REMO_IM - AttrMASK | 72.4(0.6) | 76.4(0.2) | 72.1(4.4) | 78.3(0.5) | 62.8(0.1) | 60.5(0.5) | 81.3(0.2) | 77.0(0.1) | 72.6 |
| GraphMAE | 72.0(0.6) | 75.5(0.6) | 76.3(2.4) | 83.1(0.9) | 64.1(0.3) | 60.3(1.1) | 82.3(1.2) | 77.2(0.1) | 73.9 |
| GraphMAE - USPTO | 72.8(1.1) | 75.6(0.5) | 71.8(1.1) | 83.0(0.6) | 63.8(0.3) | 58.9(0.5) | 79.5(0.5) | 78.3(0.8) | 72.97 |
| REMO_IM - GraphMAE | 71.6(0.1) | 75.8(0.1) | 77.1(2.0) | 82.1(0.2) | 64.0(0.1) | 64.3(0.2) | 85.6(1.9) | 79.8(0.1) | 75.0 |

Table 5: Results for Ablation Study of REMO_IM - AttrMASK and REMO_IM - GraphMAE on MoleculeNet.

## D  CASE LEVEL COMPARISON ON *REMO*

To gain deeper insights into the distinctions between $REMO_M$ and Graphormer-p during the pre-training phase, we perform a detailed analysis of the output logits derived from the mask reconstruction process. The prediction head, responsible for the reconstruction task, generates a vector of length 2401, which corresponds to the 2401 possible 1-hop topological combinations of atoms in the dataset. The primary objective of this prediction head is to accurately classify the masked component using the available dictionary of 2401 tokens. Through this meticulous examination, we obtain additional evidence highlighting the divergent approaches and characteristics of $REMO_M$ and Graphormer-p throughout the pre-training phase.

As depicted in Figures 9-12, the variations in logits for reconstructing the reaction centre between $REMO_M$ and Graphormer-p are visualized as square heatmaps of size $49 \times 49$, representing the 2401 possible values in the token dictionary. The target label is highlighted by red squares.

## E  ABLATION STUDY OF REMO-ATTRMASK AND REMO-GRAPHMAE

In Table 5 REMO_IM AttrMASK and REMO_IM GraphMAE undergo continuous pretraining using the updated objective function. We acknowledge that distinguishing between the performance improvement stemming from increased pretraining data and the impact of the modified objective function can be unclear. To address this concern, we have conducted comprehensive ablation studies specifically focused on REMO_IM AttrMASK and REMO_IM GraphMAE. The aim of these studies is to elucidate the precise influence of the objective function alteration.

AttrMASK - USPTO and GraphMAE - USPTO involve consistent pretraining of the original model using molecules from the USPTO dataset. Although they exhibit comparable performance to the original models across numerous benchmarks, they do not achieve comparable results to REMO for certain tasks. Remarkably, when introducing USPTO data through the original method, the overall average score even experiences a slight reduction. Nonetheless, the advantage stemming from the new objective function remains conspicuous.

## F  IMBALANCED DISTRIBUTION IN MASKED REACTION CENTRE RECONSTRUCTION

The potential challenge of imbalanced distribution in Masked Reaction Centre Reconstruction has been thoroughly examined. Specifically, we assessed the distribution of atoms and their adjacent bonds in both the ZINC dataset and the USPTO dataset, adhering to the reconstruction target construction approach outlined in Rong et al. (2020). To ensure clarity, we created a dictionary that encompasses atoms along with their neighboring bonds. This facilitated an in-depth analysis of the distribution of reconstruction targets. We conducted this analysis across three significant datasets: ZINC, the broader USPTO dataset, and the reaction centres within USPTO.

Upon analyzing the distribution of atom and edge types, we selected the top 20 labels with the highest frequencies for closer examination, addressing the issue of label imbalance.

### F.1    ZINC DATASET ATOM DISTRIBUTION

As shown in Figure 13, in the ZINC dataset, the top 20 atom labels constituted approximately 76.54% of the dataset, with the highest frequency label being *carbon connected with two aromatic bonds* at a rate of 22.58%.

### F.2    USPTO DATASET ATOM DISTRIBUTION

As shown in Figure 14, within the USPTO dataset, the top 20 atom labels accounted for around 71.53% of the dataset, and the most frequent label was *carbon connected with two aromatic bonds* at a rate of 19.79%.

### F.3    RECONSTRUCTION TARGET ATOM DISTRIBUTION

As shown in Figure 15, in the reconstruction targets of USPTO, the top 20 atom labels represented approximately 63.58% of the total reconstruction targets, with the label *nitrogen connected with a single bond* occurring most frequently at a rate of 8.62%.

It is noteworthy that the issue of imbalanced distribution exists in both the ZINC and USPTO datasets, with a prevalence of carbon atoms. However, within the context of masked reaction centre reconstruction, this concern is mitigated compared to the random masking of atoms.

Indeed, the distribution of atom types within the reconstruction target is an advantageous aspect of REMO compared to other baseline models. The reconstruction target's relative balance is a notable strength when compared to the data treatment employed by other baselines.

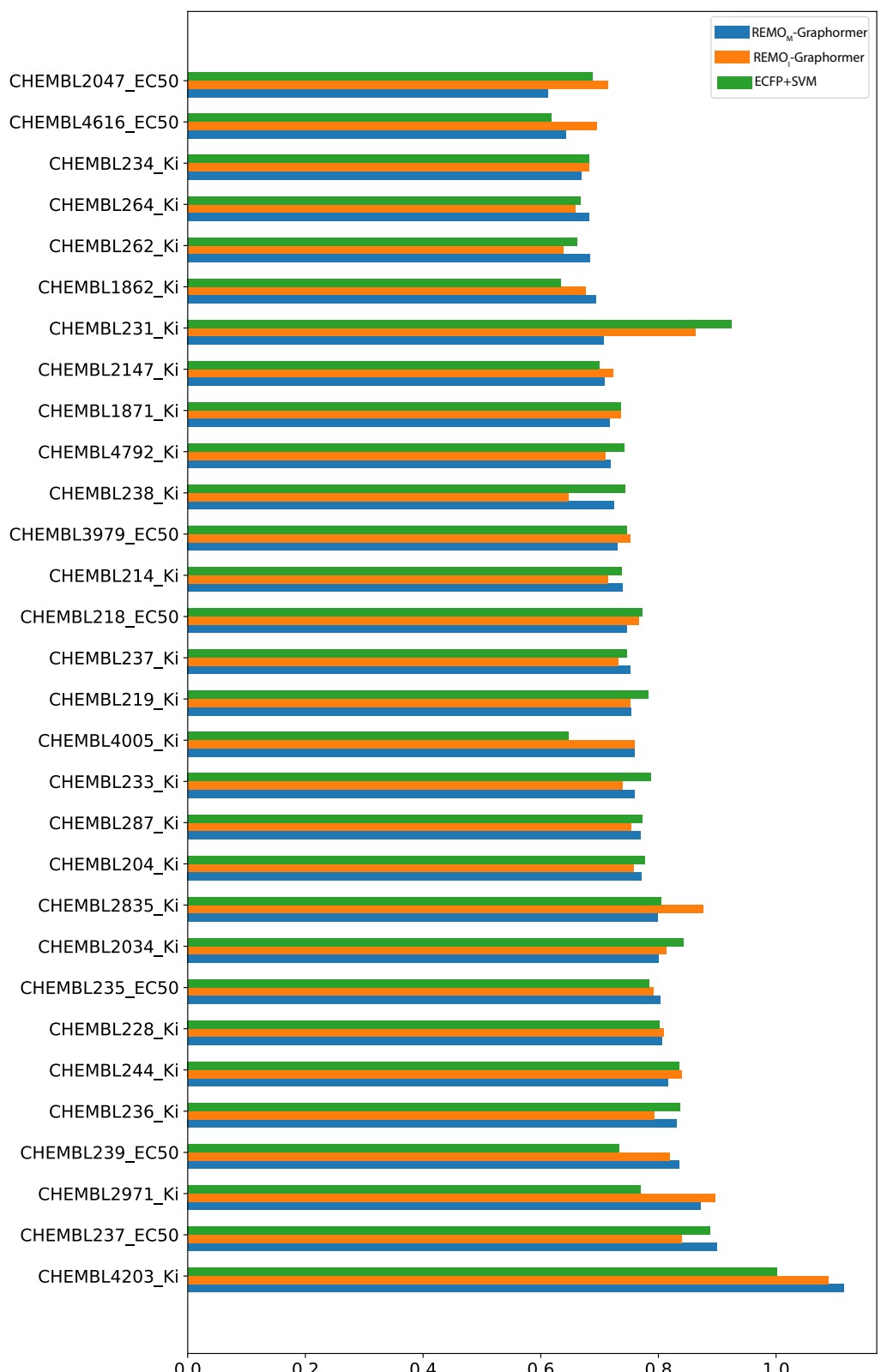

Figure 6: RMSE$_{cliff}$ of *REMO*$_\mathrm{M}$-Graphormer, *REMO*$_\mathrm{I}$-Graphormer and ECFP+SVM on MoleculeACE

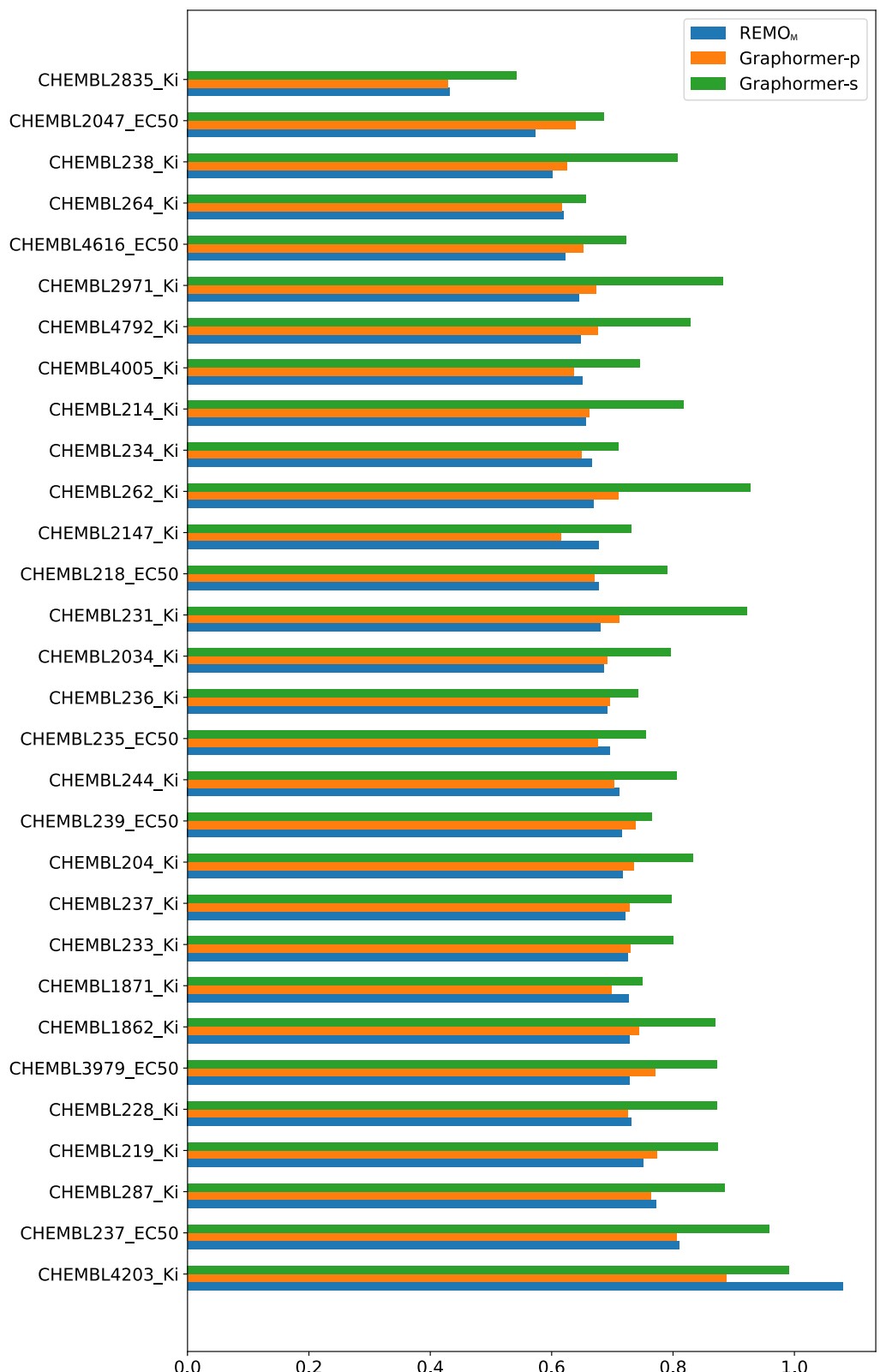

Figure 7: RMSE of $REMO_\text{M}$, Graphormer-p, and Graphormer-s on MoleculeACE

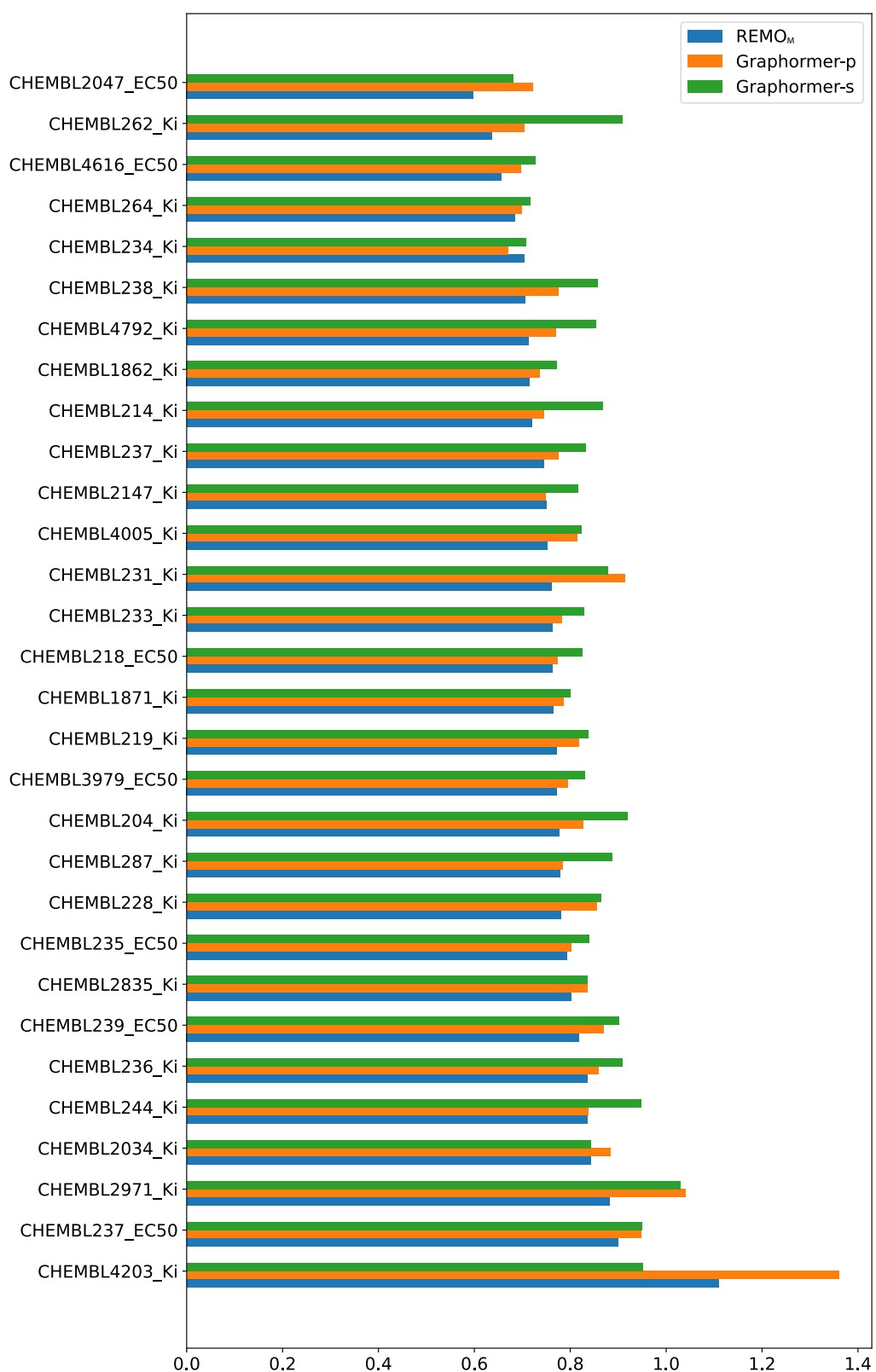

Figure 8: $\mathrm{RMSE}_{cliff}$ of $REMO_{\mathrm{M}}$, Graphormer-p, and Graphormer-s on MoleculeACE

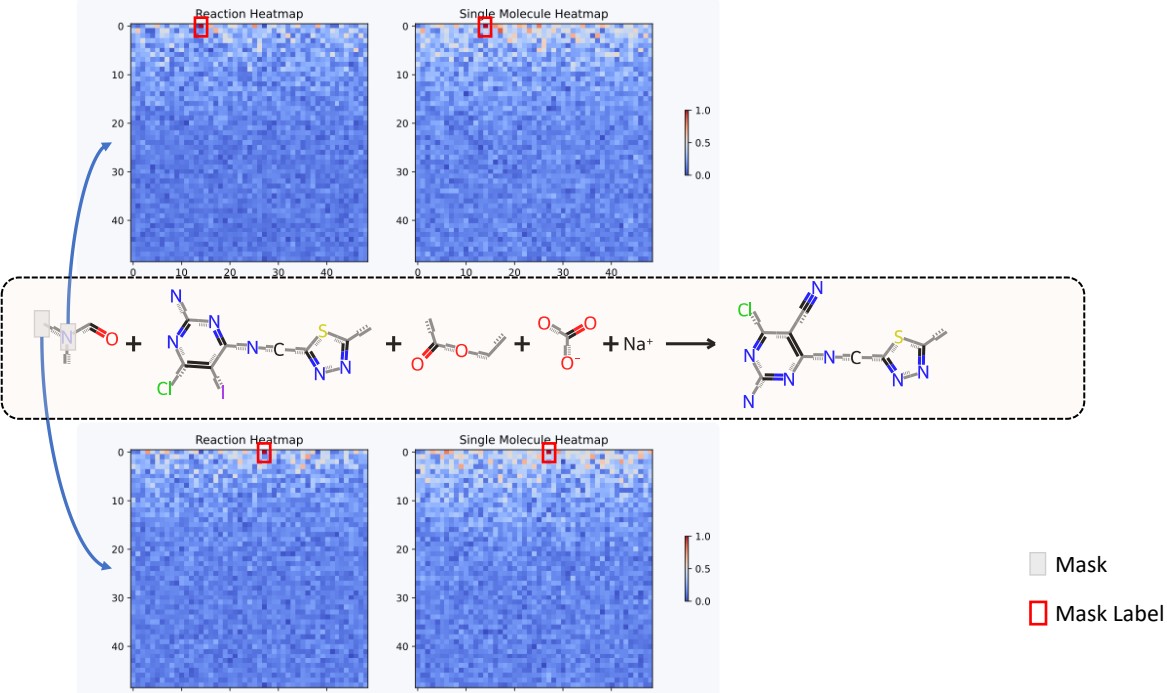

Figure 9: Case 1: Reconstruction of the reaction centre involving a carbon atom and a nitrogen atom. $REMO_M$ exhibits higher confidence in the reconstruction choice, while Graphormer-p shows increased uncertainty (evidenced by higher predicted logits for the masked nitrogen atom in indices 17 to 30, and for the masked carbon atom in indices 3 to 10).

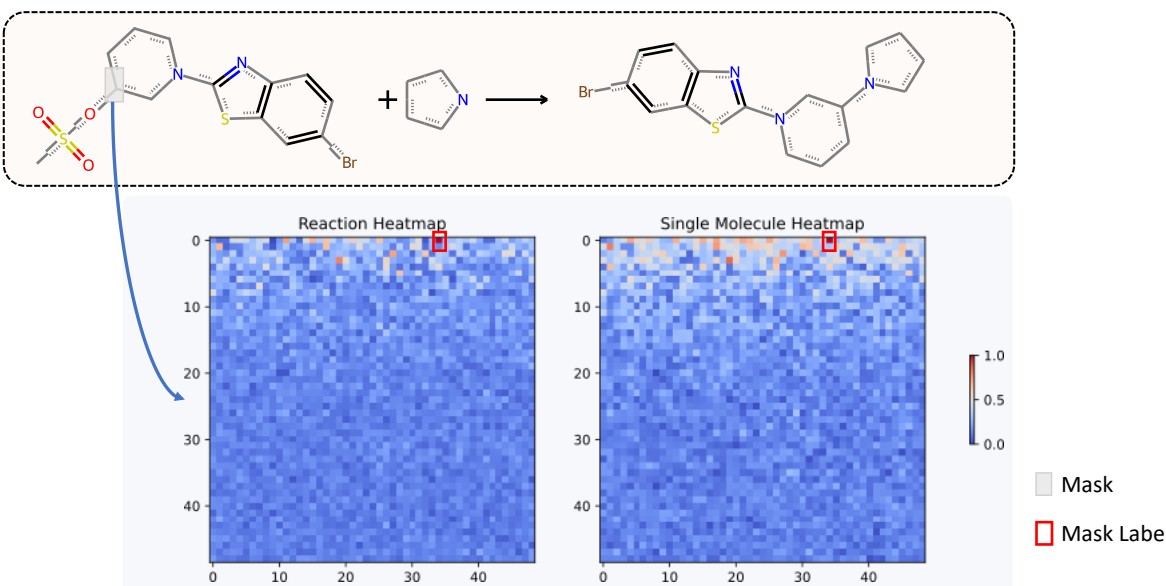

Figure 10: Case 2: Reconstruction of the reaction centre involving a carbon atom in a substituent reaction. In the absence of reaction context, Graphormer-p exhibits higher confusion, as indicated by the flatter distribution of logits observed in the heatmap.

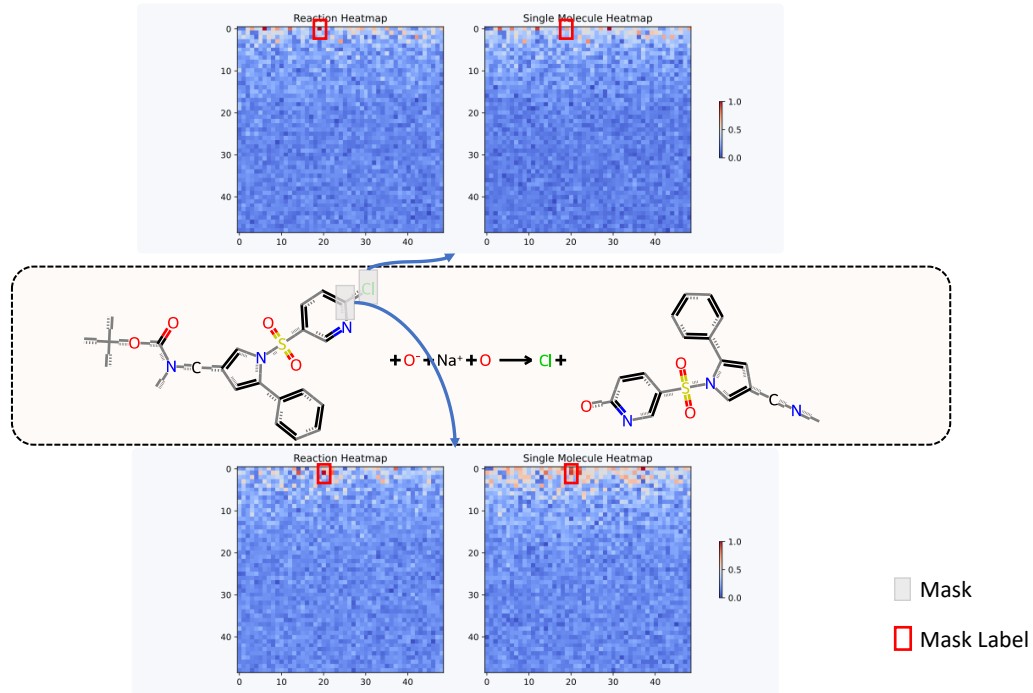

Figure 11: Case 3: Reconstruction of the reaction centre involving a carbon atom and a chlorine atom. Graphormer-p fails to reconstruct the chlorine atom in the absence of reaction context, whereas $REMO_M$ accurately identifies the correct answer. Additionally, $REMO_M$ demonstrates a more definitive choice for mask reconstruction of the carbon atom.

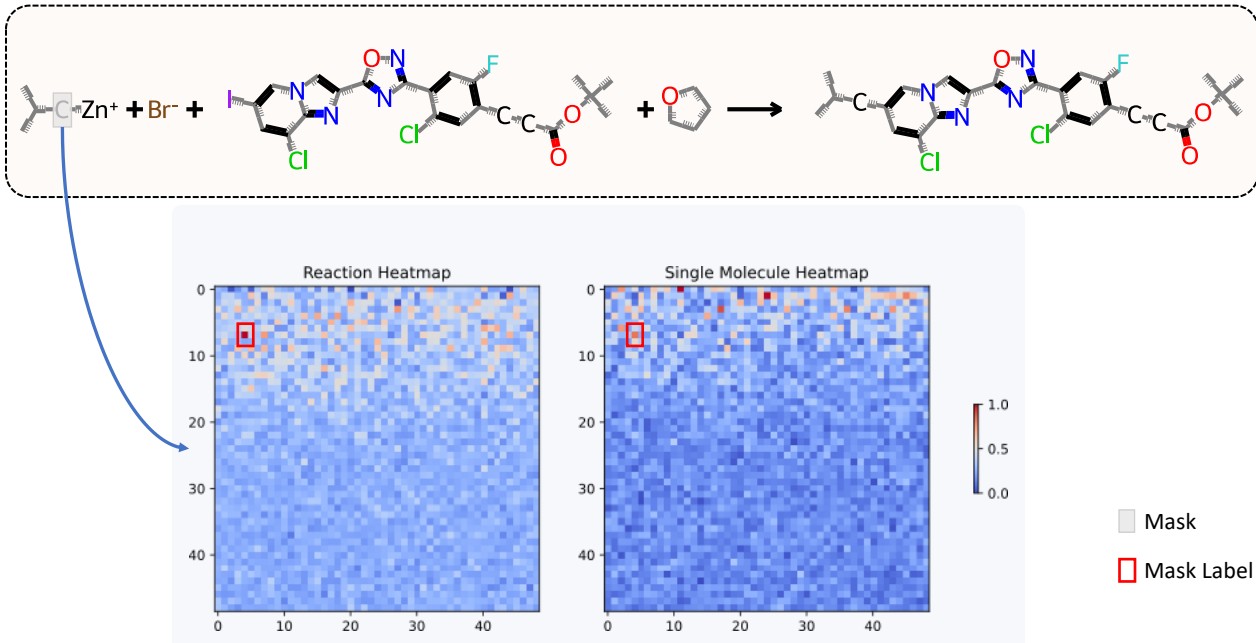

Figure 12: Case 4: The reaction involves the substitution of an iodine atom by an organozinc compound. In the absence of reaction context, Graphormer-p fails to reconstruct the reaction centre, specifically a carbon atom, whereas $REMO_M$ successfully identifies the correct reaction centre.

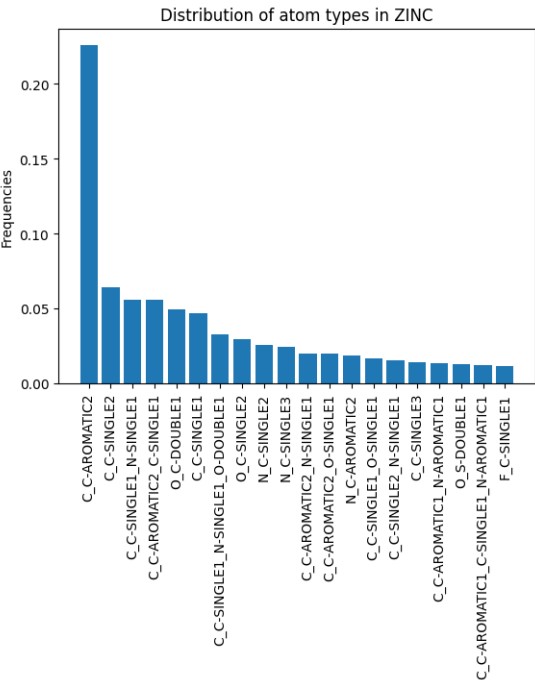

Figure 13: Distribution of Atom Types in ZINC

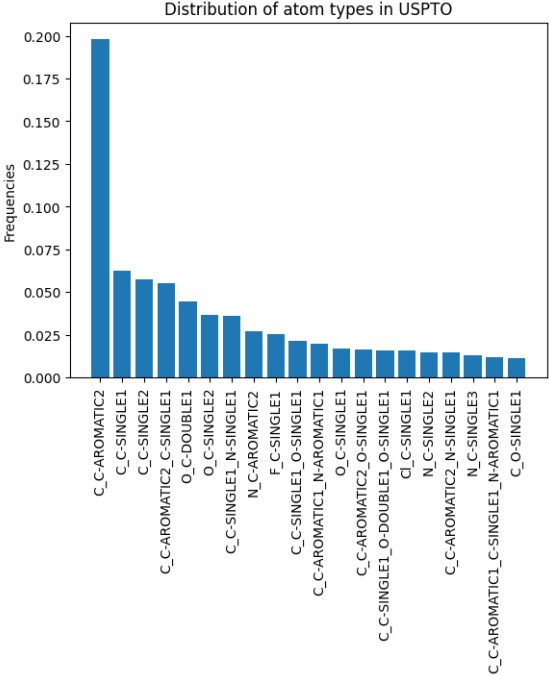

Figure 14: Distribution of Atom Types in USPTO

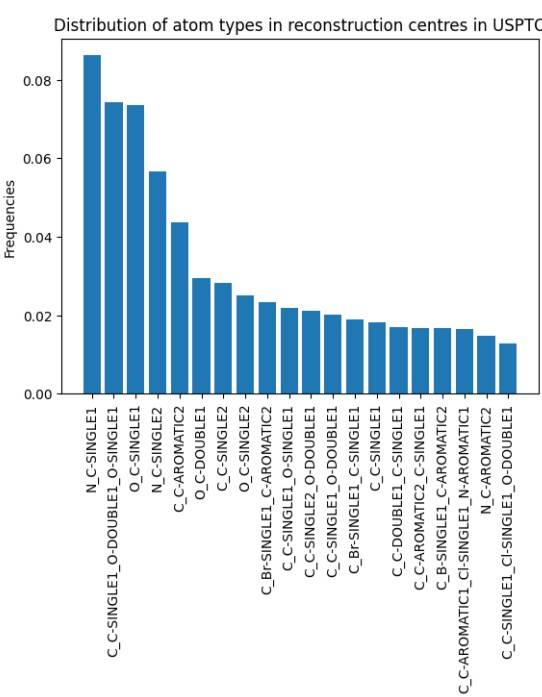

Figure 15: Distribution of Atom Types in reconstruction centres in USPTO

