# OpenReview forum: "Contextual Molecule Representation Learning from Chemical Reaction Knowledge"
_ICLR.cc/2024/Conference — Submitted to ICLR 2024_

### Official Review · Reviewer_H5fY · 2023-10-13

**Soundness:** 3 good
**Presentation:** 3 good
**Contribution:** 2 fair
**Rating:** 5
**Confidence:** 5

**Summary:**

This study focuses on understanding molecular representation by anticipating the context of hidden atoms.
While numerous prior research has delved into graph representation by reconstructing the masked components in the graph, this study seeks to reconstruct the "reaction center," believed to be vital in grasping the context of chemical reactions.
The authors start by pinpointing reaction centers as previous works - by recognizing atom pairs with differing bond types between the reactant and product graphs.
They then introduce two unique training strategies: the masked reaction center reconstruction and reaction center identification.
The former targets the prediction of the specific type of concealed atoms, while the latter determines if the atoms in a molecule belong to the reaction center.

**Strengths:**

1) Learning the representations of molecules is important for various downstream tasks.
2) Idea of identifying the reaction center is novel and the approach will definitely help the model to understand underlying chemical knowledge.
3) Extensive experiments on various downstream tasks demonstrate the superiority of REMO.

**Weaknesses:**

1) In paragraph 2 of the Introduction, the authors mentioned that "in molecule graphs the relationships between adjacent sub-units are mostly irrelevant". Is there any reference for this? I don't agree because the sub-units of the molecules are relevant to each other, and therefore, it will be helpful in reconstructing the molecule structure from the given molecule.

2) Weak experimental results.
- The most important baseline [1] is missing. It should be compared to demonstrate the effectiveness of reconstructing reaction centers instead of just learning from chemical formulas.
- In Table 1, it would be more convincing why fingerprint-based methods outperform deep models [2].
- In Table 2, comparing Mole-BERT and GraphLoG will be helpful since those methods outperform GraphMVP in Table 3.
- In Table 4, why many baseline models in previous tables are missing? Overall, various self-supervised learning methods should be compared in all tasks since they can be applied to all tasks.
- Moreover, in MolR [1], they have a task for chemical reaction classification, which aims to predict the reaction class that a chemical reaction belongs to. It would be great if REMO outperforms MolR in the task, thereby demonstrating the superiority of MolR in understanding underlying chemical reaction knowledge.

[1] CHEMICAL-REACTION-AWARE MOLECULE REPRESENTATION LEARNING, ICLR 2022.

[2] Why Deep Models Often cannot Beat Non-deep Counterparts on Molecular Property Prediction?, arxiv 2023.

3) No codes are available.

**Questions:**

Provided above.

---

> ### Author Response · Authors · 2023-11-22
> **Response to Reviewer H5fY 1/2**
>
> We sincerely appreciate your insightful comments and suggestions. We have diligently revised our manuscript in accordance with your guidance. Below, we address your raised concerns and inquiries:
>
> **On the consensus of our motivations**
>
> We apologize for any inaccuracies in our initial explanation. Our intent is to illustrate that a diverse range of atoms can form stable molecules within the realm of organic chemistry, subject to chemical laws and principles, such as the Octet Rule. This diversity surpasses that seen in the dependencies between words in sentences, given the comparatively fewer constraints. For further reference, please see:
>
> Chemistry LibreTexts. (n.d.). Combining Atoms to Make Molecules and Compounds. In Green Chemistry and the Ten Commandments of Sustainability (Manahan). Retrieved from https://chem.libretexts.org/Bookshelves/Environmental_Chemistry/Green_Chemistry_and_the_Ten_Commandments_of_Sustainability_(Manahan)/02%3A_The_Key_Role_of_Chemistry_and_Making_Chemistry_Green/2.12%3A_Combining_Atoms_to_Make_Molecules_and_Compounds
>
> **On the Concern of Weak Experiment Results**
>
> Additional experiments were conducted to **evaluate REMO_IM on the MoleculeACE benchmark **(Table 1). Our results demonstrate state-of-the-art performance, **surpassing both ECFP+SVM and other deep-learning models. The average RMSE and RMSE_Cliff are 0.647(±0.003) and 0.747(±0.004), respectively**. The full result is available in https://anonymous.4open.science/r/ICLR2024_5944-55E3/remoIM_moleculeace.csv
>
> **On the concern of lacking import baselines**
>
> We conducted a comparative analysis with MolR[1] using MoleculeNet for evaluation. Our findings indicate significant disparities between the claimed results of MolR and our own. After investigating MolR's methodology, we identified two key differences: their use of **random data partitioning** and a focus on single subtask prediction for multi-task datasets(**Please be aware that the choice of split method, whether random or strictly scaffold-based, significantly influences the performance on MoleculeNet.**). To address these differences, we concentrated on single-task datasets (HIV and BBBP) using the scaffold splitter method. Our re-evaluated results confirm REMO's superiority over MolR.
>
> | Dataset | BBBP | HIV |
> | --- | --- | --- |
> | MolR (random split, paper claimed) | 89.0 | 80.2 |
> | MolR (scaffold split) | 61.65 | 68.57 |
> | REMO (scaffold split) | 73.4 | 77.3 |
>
>
> **On More Baselines in the Benchmark ACNet**
>
> Thank you for your suggestion. We have added Mole-BERT to our set of baseline models for ACNet. Below is a detailed breakdown of Mole-BERT's performance across the 13 tasks within the ACNet Few Subset, using three different random seeds.
>
> | Models                            | AUC             | Models                           | AUC             |
> | --------------------------------- | --------------- | -------------------------------- | --------------- |
> | GROVER                        | 0.753(0.010)    | ChemBERT                      | 0.656(0.029)    |
> | MAT                            | 0.730(0.069)    | Pretrain8                     | 0.782(0.031)    |
> | pretrainGNNs                   | 0.758(0.015)    | S.T.                          | 0.822(0.022)|
> | GraphLoG                       | 0.752(0.040)    | GraphMVP                      | 0.724(0.026)    |
> | ECFP                              | 0.813(0.024)    | *REMO*                           | **0.828(0.020)**|
> | Mole-BERT| 0.758(0.014)|
>
> **On More Baselines for the Benchmark Drug-Drug Interactions (DDI)**
>
> Upon applying GraphLoG and Mol-BERT to the DDI task, it became evident that REMO outperforms these methods. This superiority can be attributed to REMO's proficiency in capturing reaction information, as it learns from reaction data as opposed to single molecules. The results align with those observed in MolNet.
>
> | Models    | f1-score    | precision    | recall  | accuracy  |
> | ----------| ------------| -------------| ------- | --------- |
> | GraphLoG  | 0.846       | 0.879        | 0.846   | 0.908     |
> | Mol-BERT  | 0.863       | 0.885        | 0.861   | 0.922     |
> | REMO_IM | 0.953 | 0.932 | 0.932  | 0.928 |
>
> [1] CHEMICAL-REACTION-AWARE MOLECULE REPRESENTATION LEARNING, ICLR 2022.

---

> ### Author Response · Authors · 2023-11-22
> **Response to Reviewer H5fY 2/2**
>
> **On the New Benchmark Reaction Type Classification**
>
> We conducted a comprehensive evaluation of REMO_IM on the Reaction Type Classification benchmark. The fine-tuning pipeline we employed aligns with the methodology used in MolR. Specifically, REMO_IM generates graph-level representations for both the reactants and products graphs, which are then concatenated to serve as input for a two-layer downstream classifier. The dataset utilized, USPTO-1k-TPL, comprises 400,000 reactions for training and 45,000 for testing, with 5% of the training set reserved for validation purposes. The results of this evaluation are as follows:
>
> | Method    | Accuracy          |
> |-----------|-------------------|
> | MolR-TAG  | 0.962     |
> | **REMO-IM** | **0.980** |
>
>
>
>
> The superior performance of REMO_IM, as evidenced by these results,  underscores its effectiveness compared MolR

---

> ### Author Response · Authors · 2023-11-22
>
> **On the Concern of code**
>
> We have restructured our code and shared it in an anonymous Git repository. Additionally, a comprehensive README file has been included, encompassing both pretraining and fine-tuning scripts, along with data processing details and the hyperparameters used during training. You can access the code repository through the following link: https://anonymous.4open.science/r/REMO-8E45/README.md.
>
> Furthermore, we have made available a pretrained model, accessible through this link: https://drive.google.com/file/d/1t62Xo5Akco9Z04El_wtbcNR_9HN0wnnf/view?usp=sharing.
>
> Should you have any questions regarding reproducibility or require further elaboration on specific details, please don't hesitate to reach out. We welcome inquiries and are ready to provide additional information as necessary.

---

> ### Author Response · Authors · 2023-11-22
>
> Dear Reviewer,
>
> We express our sincere gratitude for the invaluable suggestions you provided to enhance our manuscript. We kindly request your confirmation regarding any lingering issues that may require additional attention to meet your expectations and potentially elevate the overall assessment. Your time and feedback are highly valued, and we eagerly anticipate your response.

---

### Official Review · Reviewer_gLxF · 2023-10-27

**Soundness:** 3 good
**Presentation:** 3 good
**Contribution:** 3 good
**Rating:** 8
**Confidence:** 4

**Summary:**

The paper proposes a new self-supervised method for learning molecule representation. They propose to use masked reaction center reconstruction instead of conventional masked sub-unit reconstruction. The argument for that is in this way, the model can exploit underlying shared patterns in chemical reactions as context and can infer meaningful representations of common chemistry knowledge.

The main motivation for the paper was that  the traditional masked reconstruction loss is not enough for molecules due to:
1)  the "activity cliff" property of molecules, where a single molecular change could lead to a big difference in the molecule property, the standard self-supervised learning techniques fall short when applied to molecules.
2) BERT-like masked reconstruction loss omits the high complexity of atom combinations within molecules in nature that are quite unique compared to a simple sentence comprised of a few words.

Therefore, instead of sub-units,  they proposed to reconstruct the reaction center from the given reactants as context.
This is due to the fact that molecule biochemical or physicochemical properties are determined and demonstrated by its reaction relations to other substances.

**Strengths:**

1. The motivation/intention of the method is very clear and well-justified.
2. The paper is well-written and easy to follow.
3. The results looks good

**Weaknesses:**

The paper overall is easy to follow and easy to understand, but in terms of the baselines and experimental set up some parts could be improved. For details please reach the questions section. It would be nice if the authors add a discussion around the limitation of the proposed method.

**Questions:**

1. The graph formed part of the explanation is a bit confusing, would it be possible to make it more clear so one does not need to go back to the original paper to understand?

2. In table 1, I was wondering why REMO_IM model is not presented but only REMO_I and REMO_M? This also extended to Table 2, here the REMO represents the model trained to do the reaction center reconstruction task, the identification task, or both.

3. From Table 3, it seems often the model trained for reaction center reconstruction does not have as good performance as the one trained to predict the reaction center, any insights on this?

4. Regarding the baselines for the drug-drug interaction task, I was wondering what happens if one uses simply the graph formed without self-supervised learning, what would be the result? I think one main baseline missing from the paper here is, what happens if we do not use self-supervised learning, but directly use the proposed graph network structures to do the task, would the result be a lot worse than having the self-supervised learning setup?

5. Regarding the reaction center identification task, as the output is softmax over all the atoms, you will have a vector of [p_1, p_2,,, p_N] where 0<=p_i<=1,  what happens when you have multiple reaction center, maybe the answer to this question is very obvious but I am somehow failing to see it clearly.

---

> ### Author Response · Authors · 2023-11-22
> **Response to Reviewer gLxF**
>
> Thank you for your positive feedback. We greatly appreciate your guidance. Based on your suggestions, we have diligently revised our paper. Addressing your concerns, we provide the following responses:
>
> >Q1
>
> Our method constructs the graph through a systematic tokenization of nodes based on their types and adjacent bond types. This approach amalgamates these elements into a comprehensive dictionary of topology tokens for the dataset. For instance, a carbon atom linked with four single bonds constitutes a specific entry in this dictionary. Ultimately, our constructed dictionary encompasses approximately 2500 unique elements. The core objective of the reconstruction process is to accurately identify the corresponding label from this extensive dictionary.
>
> >Q2
>
> Additional experiments have been conducted to assess REMO_IM's performance on MoleculeACE, with comprehensive results available at https://anonymous.4open.science/r/ICLR2024_5944-55E3/remoIM_moleculeace.csv. We observed an average RMSE of 0.647(±0.003) and RMSE_Cliff of 0.747(±0.004), affirming REMO_IM's state-of-the-art (SOTA) status when benchmarked against models like ECFP+SVM and other deep-learning frameworks. To clarify any misunderstandings, Table 2 in our paper features REMO_IM, which was concurrently pre-trained on both the reaction center reconstruction and identification tasks.
>
> >Q3
>
> Table 3 highlights the distinct strengths of REMO-I and REMO-M across various tasks. REMO-I demonstrates superior performance in pharmacology-related tasks (e.g., BBBP, Tox21, Toxcast, SIDER), while REMO-M is more adept in biophysical tasks like HIV and Bace. Notably, REMO_IM, integrating both Identification and Masking pre-training, achieves the highest overall performance at 74.7, illustrating the synergistic benefits of these tasks.
>
> >Q4
>
> To demonstrate the efficacy of our pretraining techniques, we applied REMO without pretraining to the DDI task. The results, as tabulated below, clearly show that our pretraining methods substantially improve the model's performance:
>
> | Models           | f1-score | precision | recall | accuracy |
> | ---------------- | -------- | --------- | ------ | -------- |
> | REMO wo/ pretrain | 0.684    | 0.725     | 0.680  | 0.846    |
> | REMO_IM | 0.953 | 0.932 | 0.932 | 0.928 |
>
> >Q5
>
> We apologize if there was any confusion. To clarify, we represent the presence of a specific atom as a reaction center using the variable p_i where p_i = 1/0. This representation is employed for a binary classification task, distinguishing whether a given atom serves as a reaction center or not. In cases where a molecule has multiple reaction centers, the vector [p_1, p_2, …, p_N] may contain multiple instances of the value 1.
> It's important to note that we do not extend the classification further to identify different types of reaction centers. If, however, there are multiple types of reaction centers (for example, 2 types), we can modify p_i to take values in the set {1, 2, 0). In this modified representation, p_i will denote whether a specific atom is a reaction center (when p_i > 0) and, if it is, indicate the type to which it belongs.

---

### Official Review · Reviewer_rKxi · 2023-10-30

**Soundness:** 2 fair
**Presentation:** 3 good
**Contribution:** 2 fair
**Rating:** 5
**Confidence:** 4

**Summary:**

The paper proposes two novel masking approaches to pretraining on molecular data using reaction data:
(1) predict the reaction center's atom type and adjacent bonds,
(2) predict whether an atom belongs to a reaction centre.
The experiments contain different tasks, benchmarks, and focus on comparing to other masking approaches

**Strengths:**

- I agree that reaction data is a promising source which should be considered in pre-training.
- The proposed approaches are straightforward / relatively simple, but make sense.
- The experiments are nice in that they also include transformers as baselines, not just GIN, and they cover different tasks.

**Weaknesses:**

- The related work is missing all references to masking approaches in SSL beyond graphs. ICLR is a more general ML conference and graph SSL has clearly been inspired by those.
- The writing contains many statements I do not think there is clear consensus about
    - "in molecule graphs the relationships between adjacent sub-units are mostly irrelevant." - would at least need references
    - "while changing one word in a long sentence might have a relatively low impact on the semantic meaning of the full sentence," - adding "not" does not
    - "In such cases, traditional masked reconstruction loss is far from being sufficient as a learning objective." - The Molformer paper shows that simple masking can recover structure quite well if enough pre-training data is available.
    - "most biochemical or physiological properties of a molecule are determined and demonstrated by its reaction relations to other substances" - also needs references, esp. for ML readers
    - "ACNet (Zhang et al., 2023) demonstrate that existing pre-trained models are incomparable to SVM based on fingerprint on activity cliff." - I doubt that activity cliffs should be a goal / considered in pre-training. This is a particularly challenging fine-tuning scenario, I agree on that. However, in pre-training the goal is to learn a generally good embedding space which can be easily adapted in various fine-tuning scenarios. In fact, in unsupervised learning more generally (i.e., not transfer learning), a uniform space is considered as goal in many papers. It is not clear to me how an embedding space needs to look like so that it is particularly beneficial also in AC scenarios. If the authors consider those, a more detailed investigation might be useful.
- Table 1: I think the baselines are too basic and thus unrealistic, the SOTA is more advanced, e.g.
    - ECFP: might be a concatenation of ECFP+MAACs or even additional, helpful descriptors rdkit provides
    - GAT, GCN, etc. are all models from the GNN literature. There are others, e.g., D-MPNN (chemprop), which target chemical tasks.
- My current main concern is the experiment design, which is not fully clear to me.
The Table 3 comparison may be lacking. Since there are no ablation results in terms of masking, I assume the authors intended to compare this in this table as well (i.e., beyond just comparing to SOTA works in general). The baselines from related works seem to be trained on other data. However, even if these are larger datasets, they do not necessarily have to be better. In fact, USPTO contains highly diverse, special data from patents. So it is not directly clear to me that this dataset is comparable to the pre-training data the other models use. Therefore, it is not at all clear how the proposed masking actually compares to related works. It is also not clear from the table if the REMO_x models are based on GIN or Graphormer. Only if the former is the case, the comparison to most of the baselines makes sense to me, in terms of masking.

**Questions:**

see above

---

> ### Author Response · Authors · 2023-11-22
> **Response to Reviewer rKxi 1/2**
>
> Thank you for your insightful feedback. We greatly appreciate your guidance. Based on your suggestions, we have diligently revised our paper. Addressing your concerns, we provide the following responses:
>
> **On the missing references**
>
> We adhere the references to masking approaches for SSL here, and this would be in the paper of the final version:
>
> >masked language model (MLM) (Devlin et al., 2019; Lample and Conneau, 2019)
>
> >Jacob Devlin, Ming-Wei Chang, Kenton Lee, and Kristina Toutanova. 2019. BERT: Pre-training of deep bidirectional transformers for language understanding. In North American Association for Computational Linguistics (NAACL).
>
> >Guillaume Lample and Alexis Conneau. 2019. Crosslingual language model pretraining. arXiv preprint arXiv:1901.07291.
>
> **On the consensus of our motivations**
>
> "in molecule graphs the relationships between adjacent sub-units are mostly irrelevant." - We are sorry for the partially accurate information in the sentence when takes out along. We aim at pointing out that a wide variety of atoms can combine to form stable molecules under the context of organic chemistry, under the constraint of chemical laws and principles (e.g. Octet Rule). Especially, the variety is wider compares to the dependencies between words in sentences, as the constraints are less. A reference of this can be found  in:
>
> >Chemistry LibreTexts. (n.d.). Combining Atoms to Make Molecules and Compounds. In Green Chemistry and the Ten Commandments of Sustainability (Manahan). Retrieved from https://chem.libretexts.org/Bookshelves/Environmental_Chemistry/Green_Chemistry_and_the_Ten_Commandments_of_Sustainability_(Manahan)/02%3A_The_Key_Role_of_Chemistry_and_Making_Chemistry_Green/2.12%3A_Combining_Atoms_to_Make_Molecules_and_Compounds
>
> **"In such cases, traditional masked reconstruction loss is far from being sufficient as a learning objective." - The Molformer paper shows that simple masking can recover structure quite well if enough pre-training data is available.**
>
> We agree on the fact that many molecular encoder trained on masking reconstruction from single molecules could recover structures quite well. However, it could be credited by a lot of factors. Firstly, most small molecule datasets are biased, the atom types are overwhelmingly biased to carbon, as seen in Supplementary F. This imbalance in dataset make the reconstruction easier when randomly mask an atom type. Moreover, as stated in [1], the masked node prediction task on single molecules might be too easy, for the limited size of vocabulary, and the valence constraints make the prediction even easier. Moreover, as stated in the Introduction part of our paper, the model may not learn useful knowledge that could transfer to downstream tasks from simple masked reconstruction, even though they may perform good in pre-training tasks.
>
> **"most biochemical or physiological properties of a molecule are determined and demonstrated by its reaction relations to other substances" - also needs references, esp. for ML readers**
>
> Textbook such as [2] explains it well, we will include it in the final version of our paper.
>
>
> [1] Sun, R., Dai, H., & Yu, A. W. (2022). Does GNN Pretraining Help Molecular Representation? In Proceedings of the 36th Conference on Neural Information Processing Systems (NeurIPS 2022). Retrieved from https://openreview.net/pdf?id=uytgM9N0vlR#:~:text=Different%20setups%20can%20lead%20to%20opposite%20conclusions.&text=In%20conclusion%2C%20different%20from%20the,is%20effective%20in%20molecular%20domain.
>
> [2]LibreTexts. (n.d.). 13: Introduction to Biochemistry. Chemistry LibreTexts. Retrieved November 21, 2023, from https://chem.libretexts.org/

---

> ### Author Response · Authors · 2023-11-22
> **Response to Reviewer rKxi 2/2**
>
> **On Activity Cliff as a goal for pre-trained models.**
>
> Activity Cliff is defined as structural similar molecules may present significantly different binding affinities to targets. It is a good example to show the complexity for the property landscape of small molecules. Inferring from single molecules is challenging to predict activity cliff. Pre-trained model learns from single molecules, which learns a uniform space for molecular structures, though the structural-activity space is not uniform for the activity cliff problem, from a structural point of view, this discrepancy causes most pre-trained models could not yield optimised result, this problem is also studied in [3]. However, the intuition of our work aims at learning the chemical environment along with molecular structures, so the model would be aware of the molecules' position in chemical reaction space. Predicting activity cliff from structures is difficult, but activity cliff still obeys the rules in biochemistry. Intuitively, a model could be sensitive to activity cliff if it learns from the structural space and the chemical reaction space at the same time.
>
> **On baselines of MoleculeACE**
>
> The result of ECFP+MACC followed by SVM is attached at: https://anonymous.4open.science/r/ICLR2024_5944-55E3/svm_moleculeace.csv
> the average rmse and rmse_cliff is 0.658 and 0.744 respectively.
>
> The result of D-MPNN (chemprop) is attached at: https://anonymous.4open.science/r/ICLR2024_5944-55E3/chemprop_moleculeace.csv
> the average rmse and rmse_cliff is 0.750(0.002) and 0.829(0.004), respectively.
>
> it is worth noting that we evaluate our model REMO_IM on MoleculeACE after submission, the result is attached at: https://anonymous.4open.science/r/ICLR2024_5944-55E3/remoIM_moleculeace.csv
> **the average rmse and rmse_cliff is 0.647(0.003) and 0.747(0.004), respectively. The result reaches SOTA, compares to either ECFP+SVM, or (ECFP+MACCs)+SVM, or other deep-learning based models**.
>
> **On the difference in pretraining dataset**
>
> >However, even if these are larger datasets, they do not necessarily have to be better. In fact, USPTO contains highly diverse, special data from patents. So it is not directly clear to me that this dataset is comparable to the pre-training data the other models use.
>
> Undoubtedly, high-quality data can compensate for quantity limitations. However, the baseline models, which take only individual SMILES as input, may not fully exploit the wealth of chemical reaction information embedded in the USPTO data. To investigate this hypothesis, we disentangled chemical reactions into individual molecules, removing duplicates to create a dataset of approximately 2 million distinct SMILES, denoted as 'uspto-single.' Subsequently, we utilized this dataset to re-pretrain the GraphMAE for 100 epochs until the model's loss achieved convergence and stability.
>
> Following the pretraining phase, we conducted a retest using MoleculeNet and present the results in the table below.
>
> |                     | BBBP        | Tox21       | Toxcast     | Sider       | Cliontox    | MUV         | Hiv         | Bace        | Average |
> |---------------------|-------------|-------------|-------------|-------------|-------------|-------------|-------------|-------------|---------|
> | GraphMAE org + zinc | 72.0(0.006) | 75.5(0.006) | 64.1(0.003) | 60.3(0.011) | 82.3(0.012) | 76.3(0.024) | 77.2(0.010) | 83.1(0.009) | 73.8    |
> | GraphMAE + (uspto data) uspto single | 64.9(0.8)   | 75.7(0.3)   | 65.0(0.3)   | 62.5(1.4)   | 71.9(0.9)   | 75.4(2.3)   | 77.4(2.1)   | 83.1(1.3) | 72      |
>
> Notably, the performance experiences a decline when transitioning to the 'uspto-single' pretraining data. This observation suggests that the pretraining method, which is solely based on individual molecules, results in a loss of crucial chemical reaction information, leading to inferior performance. To address this limitation, there is a need to design a specialized pretraining strategy, such as REMO, to fully harness the rich information present in the USPTO data.
>
> > It is also not clear from the table if the REMO_x models are based on GIN or Graphormer.
>
> We apologize for any confusion. Specifically, the REMO_IM-GraphMAE and REMO_IM-AttrMask configurations utilize GIN as the backbone. This choice is attributed to their continuous pretraining on the encoder provided by GraphMAE or AttrMask. In contrast, all other REMO settings, with the exception of these two, employ Graphormer as the backbone.
>
> [3] Why Deep Models Often cannot Beat Non-deep Counterparts on Molecular Property Prediction?, arxiv 2023.

---

> ### Author Response · Authors · 2023-11-22
>
> Dear Reviewer,
>
> We express our sincere gratitude for the invaluable suggestions you provided to enhance our manuscript. We kindly request your confirmation regarding any lingering issues that may require additional attention to meet your expectations and potentially elevate the overall assessment. Your time and feedback are highly valued, and we eagerly anticipate your response.

---

### Official Review · Reviewer_1YRR · 2023-10-31

**Soundness:** 2 fair
**Presentation:** 2 fair
**Contribution:** 2 fair
**Rating:** 3
**Confidence:** 4

**Summary:**

This work introduces REMO, a self-supervised learning framework for MRL, which leverages well-defined rules of atom combinations in chemical reactions. REMO pre-trains graphformer encoders on a large dataset of chemical reactions and proposes two pre-training objectives: masked reaction centre reconstruction and reaction centre identification. REMO supports diverse downstream molecular tasks with minimal finetuning and outperforms traditional masked modeling approaches in various experiments.

**Strengths:**

1.	The paper is well-written and easy to follow.
2.	The experiment results are comprehensive.

**Weaknesses:**

1. My main concern revolves around the masking strategies employed and the performance of the proposed method. I find it is hard to comprehend the additional information gained from the two pre-training strategies, as they appear similar in their tasks. Besides, the inclusion of a conditional molecule as a constraint during pre-training seems necessary, yet this information is absent during the actual execution of the downstream task. This discrepancy causes a disconnect between the pre-training and downstream tasks.

2. Additionally, in Table 1, the authors conclude that their approach is inferior to ECFP+SVM. It is unclear how this supports the claim of superiority for their own method. The lack of comparative advantage raises questions about the effectiveness of their approach in relation to existing methods.

**Questions:**

1. What is the rationale behind conducting pretraining for the same task?
2. Is there a way to evaluate the significance of conditional molecule generation?
3. In Table 2, why is the prediction of reaction centers useful for cliff? Especially considering that ECFP performs better than most other pretraining methods, could we explore if other methods incorporating chemical reactions for pretraining also provide information gain?
4. Why does REMO-IM outperform REMO-IM Attrmask in Table 3?

---

> ### Author Response · Authors · 2023-11-22
> **Response to Reviewer 1YRR**
>
> We are grateful for your valuable feedback and have meticulously revised our manuscript in light of your insightful recommendations. Below, we address your concerns and queries in a detailed manner:
>
> **On the Additional Information Gain of Distinct Pre-training Strategies**
>
> We acknowledge the similarity between the tasks of Reaction Centre Identification and Masked Reaction Centre Reconstruction, but emphasize their distinct characteristics. Reaction Centre Identification involves locating the reaction center on a molecule, whereas Masked Reaction Centre Reconstruction entails rebuilding the reaction center given its location, incorporating adjacent intra-molecular information and conditional molecules. As demonstrated in Table 3 of our paper, the REMO_I (Pretrained by Identification) and REMO_M (Pretrained by Masking) models exhibit specific advantages in different sub-tasks. Most notably, REMO_IM (Pretrained by both Identification and Masking) delivers the highest overall performance, highlighting the synergistic effect of these tasks.
>
> **On the Discrepancy Between Pre-training and Downstream Tasks**
>
> During pre-training, REMO does incorporate conditional molecules, despite some downstream tasks using only single molecules. REMO's training approach is designed to concurrently encode conditional and primary molecules using a singular encoder, with weight updates driven by gradients from both task branches. This method enables REMO to learn not just the atomic combinations within a molecule, but also its broader chemical context. Consequently, REMO's graph encoder, which is effective for single molecules, remains applicable to these tasks.
>
> **On REMO's Effectiveness in Addressing Activity Cliffs**
>
> We evaluated the REMO_IM model on the MoleculeACE benchmark (as seen in Table 1), with results accessible via https://anonymous.4open.science/r/ICLR2024_5944-55E3/remoIM_moleculeace.csv. **The model's average RMSE and RMSE_cliff for the test set are 0.647(0.003) and 0.747(0.004), respectively, surpassing the ECFP+SVM method and achieving state-of-the-art performance**. While REMO-M may not universally outperform fingerprint-based models, it demonstrates improvements in 16 specific tasks (RMSE metric) and 17 tasks (RMSE_cliff metric). REMO-I, on the other hand, outperforms fingerprint-based models in 16 and 15 tasks, according to RMSE and RMSE_cliff, respectively (detailed in Supplementary C.1, Figures 5 and 6).
>
> >Q1
>
> Please refer to our explanation above regarding the additional information gain of distinct pre-training strategies.
>
> >Q2
>
> We are unclear about the context of this question as REMO does not propose any method for conditional molecule generation. Nevertheless, we acknowledge the value of this direction for future research and appreciate the suggestion.
>
> >Q3
>
> The 'Activity Cliff' concept highlights the challenge of predicting binding affinity differences in structurally similar molecules. This complexity mirrors the chemical reaction space of small molecules, as depicted in Figure 1A, where similar structures can react differently. REMO's training on chemical reaction information aims to capture this complexity, potentially aiding in understanding activity cliffs in small molecules. We believe that integrating chemical reaction knowledge into pre-training could be advantageous, and exploring this connection is a key direction for future research.
>
> >Q4
>
> The observed differences are attributable to the choice of backbone models. REMO-IM utilizes the more advanced Graphformer, while REMO-IM Attrmask employs GIN, aligning with Attrmask for a balanced comparison.

---

> ### Author Response · Authors · 2023-11-22
>
> Dear Reviewer,
>
> We extend our heartfelt appreciation for the invaluable suggestions you offered to improve our manuscript. We kindly seek your confirmation on whether any outstanding issues remain that may need further attention to meet your expectations and potentially enhance the overall assessment. Your time and feedback are greatly valued, and we eagerly await your response.

---

### Meta-Review · Area_Chair_Bu8y · 2023-12-05

**Metareview:**

The paper introduces a novel approach for using chemical reactions for representation learning in chemistry. Training general purpose models for chemical/biological applications is a very quickly developing and important area. The scarcity of meaningful tasks underscores the importance of developing new approaches for enriching representation learning in chemical applications.

The paper shows convincing evidence that chemical reaction datasets are competitive with the tested self-supervised methods and other baselines. As noted by two reviews, one of the main drawbacks of the paper is lack of clear comparison to state-of-the-art (for example UniMol and MoleBLEND on MoleculeNet benchmark) or clear ablations (perhaps most needed would be understanding to what extend this pretraining is complementary to other pretraining tasks, which could be achieved by joint pretraining with various other objectives). The authors' rebuttal has not comprehensively addressed major concerns. I recognize that it is very challenging to thoroughly compare the method due to the fragmented nature of this field. Nevertheless, it is a necessary bar to clear for acceptance because a more comprehensive comparison is necessary for the ICLR community to better understand each method’s strong and weaker sides.

All in all, I am recommending rejection at this stage. I hope that the feedback provided by the Reviewers will be helpful in improving the paper.

**Justification For Why Not Higher Score:**

More thorough comparison is missing.

**Justification For Why Not Lower Score:**

N/A

---

### Decision · Program_Chairs · 2024-01-16

Reject